# Mechanism Design via the Interim Relaxation

**Kshipra Bhawalkar**
Google Research
kshipra@google.com

**Marios Mertzanidis**
Purdue University
mmertzan@purdue.edu

**Divyarthi Mohan**
Tel Aviv University
divyarthim@tau.ac.il

**Alexandros Psomas**
Purdue University
apsomas@cs.purdue.edu

## Abstract

We study revenue maximization for agents with additive preferences, subject to downward-closed constraints on the set of feasible allocations. In seminal work, Alaei [Ala14] introduced a powerful multi-to-single agent reduction based on an ex-ante relaxation of the multi-agent problem. This reduction employs a rounding procedure which is an online contention resolution scheme (OCRS) in disguise, a now widely-used method for rounding fractional solutions in online Bayesian and stochastic optimization problems. In this paper, we leverage our vantage point, 10 years after the work of Alaei, with a rich OCRS toolkit and modern approaches to analyzing multi-agent mechanisms; we introduce a general framework for designing non-sequential and sequential multi-agent, revenue-maximizing mechanisms, capturing a wide variety of problems Alaei's framework could not address. Our framework uses an *interim* relaxation, that is rounded to a feasible mechanism using what we call a two-level OCRS, which allows for some structured dependence between the activation of its input elements. For a wide family of constraints, we can construct such schemes using existing OCRSs as a black box; for other constraints, such as knapsack, we construct such schemes from scratch. We demonstrate numerous applications of our framework, including a sequential mechanism that guarantees a $\frac{2e}{e-1} \approx 3.16$ approximation to the optimal revenue for the case of additive agents subject to matroid feasibility constraints. The simplicity of our developed two-level CRSs and OCRSs highlights the strength of our framework: even with a simple analysis, it yields state-of-the-art approximation guarantees across a wide range of settings. Finally, we show how it naturally extends to multi-parameter procurement auctions.

## 1 Introduction

We consider the problem of a revenue-maximizing seller with $m$ heterogeneous items for sale to $n$ strategic agents with additive preferences, subject to downward-closed constraints on the set of feasible allocations. Revenue maximization for multi-agent environments is a central problem in Computer Science and Economics. Beyond Myerson's [Mye81] single-item mechanism, characterizing the revenue optimal mechanism in multi-item settings is a notoriously hard problem. Revenue-optimal mechanisms are hard to compute even in basic settings, and exhibit various counter-intuitive properties [MV07, DDT13, DDT15, BCKW15, HR15, Das15, HN19, PSCW22]. An active research area strives to understand optimal and approximately optimal mechanisms from various perspectives, e.g., their computational complexity [CDW12a, CDW13a, CDW12b, CDW13b], sample complexity [CR14, HMR15, DHP16, MR16, CD17, GHZ19, GW21], robustness [BS11, CD17, DK19, LLY19, PSW19, BCD20, MMPT24], and the tradeoffs between simplicity and optimal-

ity [CHK07, CHMS10, CMS15, Yao15, RW15, CM16, CZ17, KW19, CDW19, BILW20, BGN17, KMS+19].

Influential work by Alaei [Ala14] provides a framework for constructing multi-agent mechanisms with ex-post supply constraints via a reduction to single-agent mechanism design with ex-ante supply constraints. On a high level, Alaei's framework first finds a feasible in expectation ex-ante allocation rule: a vector $x \in [0,1]^{nm}$, where $x_{i,j}$ is the probability of allocating item $j$ to agent $i$, over the randomness in the mechanism and all agents' valuations. Given this ex-ante relaxation, the framework needs a single agent mechanism for each agent $i$, such that item $j$ is allocated to agent $i$ with probability at most $x_{i,j}$. Alaei shows that running such single-agent mechanisms independently can be combined with a rounding procedure (in order to satisfy the supply constraints ex-post) to give an overall approximately optimal multi-agent mechanism. This rounding step, Alaei's solution to his "magician's problem," is an online contention resolution scheme (henceforth, OCRS), in disguise. OCRSs, later defined by Feldman et al. [FSZ21], are a widely applicable tool for rounding fractional solutions in Bayesian and stochastic online optimization problems.

In this paper, we leverage our vantage point, 10 years after the work of Alaei [Ala14], with a rich OCRS toolkit and modern approaches to analyzing multi-agent mechanisms, to introduce a novel, general framework for designing both non-sequential and sequential multi-agent, revenue-maximizing mechanisms for agents with additive preferences, subject to downward-closed constraints on the set of feasible allocations. Our framework uses an *interim* relaxation, that is rounded to a feasible mechanism, using what we call a two-level OCRS, allowing for some structured dependence between the activation of its input elements. For a wide family of constraints, we can construct such schemes using OCRSs as a black box; for other constraints, e.g., knapsack, we construct such schemes from scratch. We demonstrate numerous applications of our framework, including a sequential mechanism that guarantees a $\frac{2e}{e-1} \approx 3.16$ approximation to the optimal revenue for the case of additive agents subject to matroid feasibility constraints. We also show how our framework can be easily extended to multi-parameter procurement auctions, where we provide an OCRS for Stochastic Knapsack that might be of independent interest.

## 1.1 Our Contributions

Our framework relies on an interim relaxation. Intuitively, an interim form (or reduced form) of a mechanism $\mathcal{M}$ has variables $\pi_{i,j}^{\mathcal{M}}(v_i)$, which indicate the probability that agent $i$ receives item $j$ when reporting valuation $v_i$ to $\mathcal{M}$ (over the randomness in $\mathcal{M}$, as well as the randomness in other agents' valuations), and variables $q_i^{\mathcal{M}}(v_i)$, which indicate the expected payment of agent $i$ when reporting valuation $v_i$ to $\mathcal{M}$ (over the same randomness). Writing a linear program that optimizes revenue over the space of all feasible interim rules has proven to be a useful endeavor when computing optimal and approximately optimal mechanisms [CDW12a, CDW13a, CDW12b, CDW13b], as well as for deriving upper bounds (via duality) to the revenue optimal mechanism [CDW19]. While the number of variables in this program (corresponding to interim rules) is polynomial, the number of constraints needed to ensure that an interim rule is feasible (i.e., that there exists a mechanism $\mathcal{M}$ that induces it) is typically exponential (even for, e.g., the simple case of selling a single item to $n$ agents). Our starting point is to consider a relaxation of these feasibility constraints, resulting in interim rules that are *feasible in expectation*.

Given (optimal or approximately optimal) interim rules that are feasible in expectation the first natural step for rounding to an actual mechanism is to use a CRS/OCRS. Contention resolution schemes, or CRSs, were defined by Chekuri et al. [CVZ14] as a tool for rounding fractional solutions in (submodular) optimization problems. In this framework, there is a finite ground set of elements $N = \{e_1, \ldots, e_k\}$, a downward-closed family $\mathcal{F}$ of subsets of $N$, and a fractionally feasible point $x^* \in [0,1]^k$. The main idea is to obtain a (possibly infeasible) random set $R(x^*)$ from $x^*$, by treating $x^*$ as a product distribution: element $i$ is included in $R(x^*)$ with probability $x_i^*$. Given $R(x^*)$, a $c$-selectable CRS selects a set $I \subseteq R(x^*)$ that is feasible (i.e., $I \in \mathcal{F}$), in a way that each element is selected with probability at least $c$ if it is in $R(x^*)$. A refined definition, $(b,c)$-selectable CRSs, for $b \in [0,1]$, extends the concept of $c$-selectable CRSs (which are simply $(1,c)$-selectable) and provides the same guarantee per element when given as input a set $R(b\,x^*)$. Feldman et al. [FSZ21] extended this framework to online settings and defined OCRSs.

Back to our problem, a first blueprint for a mechanism, given interim rules that are feasible in expectation, would be to elicit reports $r_1, \ldots, r_n$ from the agents, and then construct a set of active

elements (the input to the CRS/OCRS) according to $\pi_{i,j}(r_i)$, for every agent $i$ and item $j$. A technical complication is that CRSs/OCRSs receive as input elements that become active *independently*. In our blueprint, the event that element "$(i, j, r_i)$" becomes active is correlated with the event that element "$(i, j', r_i)$" becomes active. To bypass this obstacle, we define variants of CRSs and OCRSs which we call *two-level contention resolution schemes*, or tCRS, and *two-level online contention resolution schemes*, or tOCRS. Intuitively, a tCRS/tOCRS receives an $n$ by $m$ matrix of elements, such that elements in the same row are independent, conditioned on the value of a row-specific random variable (and these row-specific random variables are independent).

**Our framework.** Informally, our overall framework takes as input feasible in expectation interim rules and a tCRS/tOCRS and outputs a mechanism; for the case of tOCRS the mechanism is sequential, i.e., it sequentially approaches each agent $i$, elicits a report $r_i$, and decides on the outcome of agent $i$ (her allocation and payment) before proceeding to the next agent. We require that valuations are independent across agents, but the value of agent $i$ for item $j$ can be correlated with her value for item $j'$. Our mechanisms are Bayesian incentive compatible (BIC) and Bayesian individually rational (BIR). Given an $\alpha \geq 1$ approximately optimal interim form, and a $(b, c)$-selectable tCRS/tOCRS, our mechanism is $\frac{\alpha}{b\,c}$ approximately optimal (Theorem 1). See Figure 1.

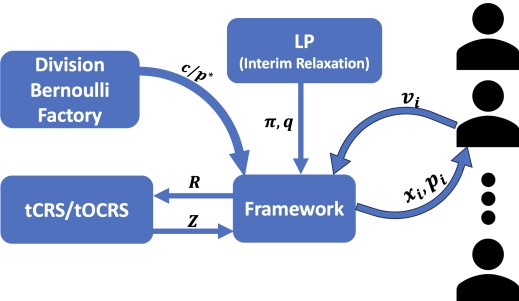

Figure 1: Given an interim form $(\pi, q)$ our framework (sequentially in the case of tOCRSs) elicits agents' valuations. Given the report $v_i$ of agent $i$, it executes the tCRS/tOCRS on a set of active elements $R$, and returns a set $Z$. The allocation of $i$ is constructed from $Z$; when given only black-box access to the tCRS/tOCRS this construction can be done efficiently via a division Bernoulli factory.

Towards constructing tCRSs/tOCRSs, we first give a general reduction for constructing a tCRS/tOCRS, that uses CRSs/OCRSs as a black-box (Theorem 2). Informally, if CRSs/OCRSs for certain feasibility sets $\mathcal{F}_1, \ldots, \mathcal{F}_k$ exist, we can provide a tCRS/tOCRS for certain combinations of the $\mathcal{F}_i$s. Combining with known CRS/OCRS results we get tCRSs/tOCRSs for various settings of interest. Next, we give $1/10$-selectable tOCRS for Knapsack constraints and a $1/9$-selectable tOCRS for Multi-Choice Knapsack constraints (Theorems 3 and 4). Notably, our tOCRS for knapsack implies a $1/10$-selectable OCRS for knapsack, which is better than the $0.085$-selectable OCRS given by Feldman et al. [FSZ21], but not as good as the state-of-the-art $1/(3 + e^{-2})$-selectable ($\approx 0.319$-selectable) OCRS of Jiang et al. [JMZ22].

**Applications.** Plugging the aforementioned tCRSs/tOCRSs into our framework gives numerous interesting applications. As a first application, consider the problem of auctioning off $m$ items to $n$ agents with additive preferences, such that the set of agents that each item $j$ is allocated to must be an independent set of a matroid, and the items allocated to each agent must be an independent set of a matroid. Our results give a sequential, BIC and BIR mechanism that guarantees a $\frac{2e}{e-1} \approx 3.16$ approximation to the optimal revenue (Application 1). The previously best-known approximation possible by a sequential mechanism was 70 (for the special case where every item can be allocated at most once, and where items' values are independent), due to Cai and Zhao [CZ17], whose mechanism has additional desirable simplicity properties (that we do not guarantee here, and are in fact impossible to guarantee for correlated items [BCKW15, HN19]). Beyond its simplicity, their mechanism applies to a broader class of valuation functions.

As a next application, consider the problem of auctioning off $m$ items to $n$ agents with additive preferences, where each item $j \in [m]$ has some weight $k_j$ and the total weight of items sold cannot exceed $K$. Our results imply that there exists an efficiently computable, sequential, BIC and BIR mechanism that guarantees a 10 approximation to the optimal revenue. Additionally, if each agent $i$

can get at most one item, there exists an efficiently computable, sequential, BIC and BIR mechanism that guarantees a 9 approximation to the optimal revenue (Application 2). Finally, consider the problem of auctioning off $m$ items to $n$ agents with *arbitrary* valuation functions, where each item $j \in [m]$ has some weight $k_j$ and the total weight of items sold cannot exceed $K$. Then, our results imply that there exists a (computationally inefficient) sequential, BIC and BIR mechanism that guarantees a 9 approximation to the optimal revenue (Application 3).

**Extensions.** Our framework can be easily extended to other mechanism design problems, beyond auctioning off items to agents. In Appendix F we give an extension to *procurement auctions*, where a value-maximizing buyer is interested in buying services from strategic sellers, subject to a budget constraint. In this case, we show how, given an OCRS for Stochastic Knapsack (see Appendix F for definitions) it is possible to design an approximately optimal sequential multi-parameter procurement auction. Combining with a result of Jiang et al. [JMZ22], we then have a $(3 + e^{-2})$-approximately optimal sequential procurement auction (Application 4). As an aside, we also give an OCRS for the stochastic knapsack setting that might be of independent interest (Theorem 6). Feldman et al. [FSZ21] give a greedy and monotone[1] $(3/2 - \sqrt{2})$-selectable OCRS ($\approx 0.0858$) for Knapsack, while Jiang et al. [JMZ22] give a $\frac{1}{3+e^{-2}}$-selectable OCRS ($\approx 0.319$) for Stochastic Knapsack (this OCRS induces a non-greedy and non-monotone OCRS for Knapsack). We give $c$-selectable OCRS for Stochastic Knapsack (that induces a greedy and monotone OCRS for Knapsack), where $c = \max\{\frac{1-k^*}{2-k^*}, 1/6\}$, and $k^*$ is a parameter that depends on the maximum possible weight (in the support of the distributions from which the stochastic weights are drawn); our OCRS is therefore always better than the OCRS of Feldman et. al., and better than the OCRS of Jiang et. al. when $k^*$ is small.

## 2 Preliminaries

We consider the problem of a seller with $m$ indivisible, heterogeneous items for sale to $n$ strategic agents. Each agent $i$ has a private valuation vector $v_i$ that is drawn independently from an $m$-dimensional distribution $\mathcal{D}_i$ (that is known to the seller). We write $\mathcal{V}_i = supp(\mathcal{D}_i)$ for the set of possible valuations for agent $i$. Agent $i$ has a value $v_{i,j}$ for item $j$. We write $\mathcal{D}_{i,j}$ for the marginal distribution for item $j$, noting that $\mathcal{D}_{i,j}$ is not necessarily independent of $\mathcal{D}_{i,j'}$. We assume that agents have *additive preferences*, i.e., the value of agent $i$ with valuation $v_i$ for a subset of items $S \subseteq [m]$ is $\sum_{j \in S} v_{i,j}$. Agents are *quasi-linear*: the utility of an agent is her value minus her payment. An (integral) allocation $x \in \{0,1\}^{n \cdot m}$ indicates which item was received by which agent, i.e., $x_{i,j} \in \{0,1\}$ is the indicator for whether agent $i$ received item $j$. There are constraints on the set of feasible allocations represented by a set $\mathcal{F} \subseteq \{0,1\}^{n \cdot m}$; that is, an allocation $x$ is feasible if $x \in \mathcal{F}$ (therefore, one can equivalently think of the agents as constrained additive). Let $P_{\mathcal{F}}$ be the convex hull of all characteristic vectors of $\mathcal{F}$, i.e. $P_{\mathcal{F}} = conv\{\mathbf{1_F} : F \in \mathcal{F}\}$. We write $P_{\mathcal{F}}^i$ for the polytope that corresponds to agent $i$, i.e., the polytope $P_{\mathcal{F}}$ when we only consider the $m$ dimensions that correspond to the allocation of agent $i$.

### 2.1 Mechanism Design Preliminaries

A mechanism $\mathcal{M}$ takes as input a reported valuation from each agent and selects a (possibly random) allocation in $\mathcal{F}$, and payments to charge the agents. An agent's objective is to maximize her expected utility. A mechanism $\mathcal{M}$ is *Bayesian Incentive Compatible (BIC)* if every agent $i \in [n]$ maximizes her expected utility by reporting her true valuation $v_i$, assuming other agents do so as well, where this expectation is over the randomness of other agents' valuations, as well as the randomness of the mechanism. A mechanism is *Bayesian Individually Rational* (BIR) if every agent $i \in [n]$ has non-negative expected utility when reporting her true valuation (assuming other agents do so as well). The (expected) revenue of a BIC mechanism is the expected sum of payments made when agents draw their valuations from $\mathcal{D}$ (and report their true valuations to the mechanism). We say that a mechanism is BIC-IR if it is both BIC and BIR.

---

[1]An OCRS is greedy if it fixes a downward-closed family of feasible sets before the (online) process starts, and greedily accepts any active element $e$ that will not violate feasibility if included. An OCRS $\mu$ is *monotone* if for all $e \in A \subseteq B$, the probability that $\mu$ selects $e$ when $A$ is the set of active elements is at most the probability that $\mu$ selects $e$ when $B$ is the set of active elements. These properties are important for applications in submodular optimization [CVZ14].

The *optimal mechanism* for a given distribution $\mathcal{D}$, whose revenue is denoted by $\mathrm{Rev}(\mathcal{D})$, maximizes expected revenue over all BIC-IR mechanisms. For a given mechanism $\mathcal{M}$, we slightly abuse notation and write $\mathrm{Rev}^{\mathcal{M}}(\mathcal{D})$ to denote its revenue under a distribution $\mathcal{D}$. A mechanism guarantees an $\alpha$ approximation to the optimal revenue if $\alpha \, \mathrm{Rev}^{\mathcal{M}}(\mathcal{D}) \geq \mathrm{Rev}(\mathcal{D})$. Finally, we say that a mechanism is *sequential* if it sequentially approaches agent $i$, elicits a report $r_i$, and allocates items to $i$ before proceeding to the next agent.

**Interim allocations and payments.** The *interim allocation* of a mechanism $\mathcal{M}$, $\pi^{\mathcal{M}}$, indicates, for each agent $i$ and item $j$ the probability $\pi_{i,j}^{\mathcal{M}}(r_i)$ that agent $i$ receives item $j$ when she reports valuation $r_i$ (over the randomness in $\mathcal{M}$ and the randomness in other agents' reported valuations $v_{-i}$, drawn from $\mathcal{D}_{-i}$). The *interim payment* of agent $i$, $q_i^{\mathcal{M}}(r_i)$, is the expected payment she makes when she reports valuation $r_i$ (again, over the randomness in $\mathcal{M}$ and the randomness in other agents' reported valuations). It is easy to see that the expected utility of agent $i$ with valuation $v_i$ when reporting $r_i$ to a mechanism $\mathcal{M}$, is $\sum_{j \in [m]} v_{i,j} \pi_{i,j}^{\mathcal{M}}(r_i) - q_i^{\mathcal{M}}(r_i)$. We will drop the superscript $\mathcal{M}$ when the mechanism is clear from the context.

Given interim allocations and payments, it is not a straightforward task to determine whether they are *ex-post feasible*, i.e., whether there exists a mechanism $\mathcal{M}$ that induces the exact probabilities promised by the interim allocations. In fact, doing this task efficiently is at the core of the framework of Cai et al. [CDW12a, CDW13a, CDW12b, CDW13b] for computing approximately optimal mechanisms. However, it is typically straightforward to find interim allocations that are *feasible in expectation*.

**Definition 1** (Feasibility in expectation). *An interim allocation rule $\pi$ is feasible in expectation if (i) $\forall i \in [n], v_i \in \mathcal{V}_i, \pi_i(v_i) \in P_{\mathcal{F}}^i$, and (ii) $\left[ \sum_{v_i \in \mathcal{V}_i} \Pr[v_i] \cdot \pi_{i,j}(v_i) \right]_{(i,j) \in [n] \times [m]} \in P_{\mathcal{F}}$.*

We say that an interim allocation, payment pair $(\pi, q)$ is BIC if $\forall i \in [n], v_i, v_i' \in \mathcal{V}_i$ it holds that

$$\sum_{j \in [m]} v_{i,j} \pi_{i,j}(v_i) - q_i(v_i) \geq \sum_{j \in [m]} v_{i,j} \pi_{i,j}(v_i') - q_i(v_i').$$

An interim allocation, payment pair $(\pi, q)$ is BIR if $\forall i \in [n], v_i \in \mathcal{V}_i, \sum_{j \in [m]} v_{i,j} \pi_{i,j}(v_i) - q_i(v_i) \geq 0$. An interim allocation, payment pair $(\pi, q)$ is BIC-IR if it is both BIC and BIR. Finally, an interim allocation, payment pair $(\pi, q)$ guarantees an $\alpha$-approximation to the optimal revenue if $\alpha \left( \sum_{i \in [n]} \sum_{v_i \in \mathcal{V}_i} \Pr[v_i] \cdot q_i(v_i) \right) \geq \mathrm{Rev}(\mathcal{D})$.

## 2.2 OCRS and tOCRS Preliminaries

Consider a finite ground set $N = \{e_1, \cdots, e_k\}$ and a family of feasible subsets $\mathcal{F} \subseteq 2^N$. Let $P_{\mathcal{F}} = conv(\{\mathbf{1}_I | I \in \mathcal{F}\})$ be the convex hull of all characteristic vectors of feasible sets. Let $x \in P_{\mathcal{F}}$, and let $R(x) \subseteq N$ be a random set obtained by including each element $i \in N$ independently with probability $x_i$. The set $R(x)$ is feasible in expectation (with respect to $\mathcal{F}$) but not necessarily ex-post feasible. Given $R(x)$, a contention resolution scheme (CRS) selects a subset $I \subseteq R(x)$ such that $I \in \mathcal{F}$. If elements of $R(x)$ are given in an online manner, the corresponding scheme is called an online contention resolution scheme (OCRS). To avoid trivial solutions (e.g., $I = \emptyset$), we would additionally like to have the property that each element $i \in N$ appears in $I$ with probability at least $cx_i$ for some $c$. We call such schemes $c$-selectable. Some schemes only work if elements come from $R(b\,x)$; such schemes are called $(b, c)$-selectable. Formally:

**Definition 2** (Online Contention Resolution Scheme(OCRS) [FSZ21]). *Let $b, c \in [0, 1]$. For every $x \in P_{\mathcal{F}}$, let $R(b\,x)$ be a random subset of active elements, where element $i \in N$ is active with probability $b\,x_i$, independently. A $(b, c)$-selectable Online Contention Resolution scheme (OCRS) $\mu$ for $P_{\mathcal{F}}$ is a (possibly randomized) online procedure that, given active elements one by one, decides whether to select an active element irrevocably before the next element is revealed. The OCRS $\mu$ returns a set $I \subseteq R(b\,x)$, such that (i) $I \in \mathcal{F}$, and (ii) $\Pr\left[i \in I | i \in R(b\,x)\right] \geq c, \forall i \in N$.*

We introduce a variant of the previous OCRS model that allows for dependencies between the activation of different elements. This slightly changes the setup, as well as the definition of a "scheme." Consider the ground set $N = \{e_{i,j}\}_{i \in [n], j \in [m]}$, where $|N| = n\,m$, and a family of feasible subsets $\mathcal{F} \subseteq 2^N$. Let $P_{\mathcal{F}} = conv(\{\mathbf{1}_I | I \in \mathcal{F}\})$ be the convex hull of all characteristic vectors

of feasible sets. Let $P_{\mathcal{F}}^i$ be the restriction of $P_{\mathcal{F}}$ to the $m$ dimensions that correspond to elements $(i,j)$, $j \in [m]$. Elements will become active in a certain, dependent way, as induced by a *two-level stochastic process* $(\mathcal{D}, x)$, defined as follows:

**Definition 3** (Two-Level stochastic process). *We say that $(\mathcal{D}, x)$ is a two-level stochastic process over $\{0,1\}^{n\,m}$, where $\mathcal{D} = \times_{i=1}^n \mathcal{D}_i$ is a product distribution and $x \in [0,1]^{m \sum_{i=1}^n |\mathcal{V}_i|}$, if it is induced by the following procedure: (i) we first sample $d_i$ from $\mathcal{D}_i$, independently, and (ii) for each $(i,j) \in [n] \times [m]$, element $e_{i,j}$ becomes active with probability $x_{i,j}(d_i)$, independently.*

For the case of OCRSs, since elements became active independently, (expected) feasibility for a product distribution $x$ boiled down to $x$ being fractionally feasible for $\mathcal{F}$, i.e. $x \in P_{\mathcal{F}}$. Here, since elements are active in a dependent way, our notion of feasibility needs to be further refined:

**Definition 4** (Feasibility). *Let $\mathcal{F} \subseteq 2^{[n] \times [m]}$ be a feasibility set and $P_{\mathcal{F}}$ be its relaxation. We say that a two-level stochastic process $(\mathcal{D}, x)$ is feasible with respect to $\mathcal{F}$ if:*

1. *For each $i \in [n]$ and each $d_i \in supp(\mathcal{D}_i)$, $(x_{i,1}(d_i), \cdots, x_{i,m}(d_i)) \in P_{\mathcal{F}}^i$.*

2. *$w \in P_{\mathcal{F}}$, where $w_{i,j} = \sum_{d_i \in supp(\mathcal{D}_i)} \Pr[\mathcal{D}_i = d_i]\, x_{i,j}(d_i)$.*

Let $(\mathcal{D}, x)$ be a two-level stochastic process that is feasible with respect to $\mathcal{F}$, and let $R(\mathcal{D}, x) \subseteq N$ be a random set of elements obtained by sampling from $(\mathcal{D}, x)$. Our goal is to select a subset $I \subseteq R(\mathcal{D}, x)$ (possibly online) such that $I \in \mathcal{F}$ and the probability that an active element is selected is lower bounded by a constant $c$. Formally, our definitions of two-level Online Contention Resolution Schemes (tOCRSs) is as follows.

**Definition 5** (Two-level OCRS (tOCRS)). *Let $b, c \in [0,1]$. Let $(\mathcal{D}, x)$ be a two-level stochastic process that is feasible with respect to $\mathcal{F}$, and let $R(\mathcal{D}, b\,x) \subseteq N$ be a random set of elements obtained by sampling from $(\mathcal{D}, b\,x)$. Elements of $R(\mathcal{D}, b\,x)$, and the corresponding samples from the first level of $(\mathcal{D}, x)$, appear online, in batches of size $m$: the process selects some $i \in [n]$ and reveals $d_i$ (sampled from $\mathcal{D}_i$) and all elements $\{e_{i,j}\}_{j \in [m]}$, before selecting a new $i' \in [n]$. A $(b,c)$-selectable two-level OCRS (tOCRS) $\mu$ for $\mathcal{F}$ is a (possibly randomized) online procedure that, given active elements that satisfy the aforementioned ordering, decides whether to select an active element irrevocably before the next element is revealed, i.e., returns a set $I \subseteq R(\mathcal{D}, b\,x)$, such that (i) $I \in \mathcal{F}$, and (ii) $\Pr[i \in I | i \in R(\mathcal{D}, b\,x)] \geq c, \forall i \in N$.*

The existence of a $(b, c)$-selectable tOCRS implies the existence of a $(b, c)$-selectable OCRS, by a simple simulation argument (for $m = 1$).

Due to space constraints, preliminaries on CRSs and tCRSs are deferred to Appendix B.1, and preliminaries on Bernoulli factories are deferred to Appendix B.2.

## 3 Mechanisms from two-level OCRSs

We give two frameworks for constructing mechanisms. Due to space limitations the framework for tCRSs can be found in Appendix C. The input to a framework is (i) a BIC-IR interim allocation, payment rule pair $(\pi, q)$ that is feasible in expectation and is an $\alpha$ approximation to the optimal revenue, (ii) a $(b, c)$-selectable tCRS/tOCRS for $\mathcal{F}$, and (iii) a parameter $\epsilon \geq 0$. Our frameworks produce BIC-IR, $\frac{\alpha}{b(c-\epsilon)}$ approximately optimal (and sequential, for tOCRSs) mechanisms for agents with constrained (with respect to $\mathcal{F}$) additive valuations.

Given a tOCRS, our framework, Algorithm 1, works as follows. We approach each agent $i$ sequentially. Agent $i$ reports $r_i^* \in \mathcal{V}_i$ and pays $b(c-\epsilon) \cdot q_i(r_i^*)$. We consider each item $j$ sequentially. We make element $(i,j)$ active with probability $b \cdot \pi_{i,j}(r_i^*)$, and then ask the tOCRS if this element should be selected (assuming it was active). If the tOCRS selects the element $(i,j)$, we again flip an additional coin to decide whether agent $i$ should get item $j$. This last coin essentially balances the randomness of the tOCRS and ensures that the probability that agent $i$ gets item $j$ when they report $r_i^*$ is exactly $b(c-\epsilon)\pi_{i,j}(r_i^*)$, which, combined with the chosen payment, ensures that BIC and BIR are satisfied.

**Theorem 1.** *Given (i) a BIC-IR interim allocation, payment rule pair $(\pi, q)$ that is feasible in expectation and is an $\alpha \geq 1$ approximation to the optimal revenue (ii) a $(b, c)$-selectable tOCRS (resp. tCRS) for $\mathcal{F}$, and (iii) a parameter $\epsilon \geq 0$, Algorithm 3 and Algorithm 1 give BIC-IR, $\frac{\alpha}{b(c-\epsilon)}$-approximately optimal (and sequential for the case of Algorithm 1) mechanisms for $\mathcal{F}$.*

---
**ALGORITHM 1:** Our framework for tOCRSs

**Input:** allocation, payment rule pair $(\pi, q)$, $(b, c)$-selectable tOCRS $\mu$, parameter $\epsilon \geq 0$.

---
**for** *each agent* $i \in [n]$ **do**
    Agent $i$ reports $r_i^* \in \mathcal{V}_i$ and pays $b\,(c - \epsilon)\,q_i(r_i^*)$.
    **for** *each item* $j \in [m]$ **do**
        $R_{i,j} \leftarrow 1$ with probability $b\,\pi_{i,j}(r_i^*)$.
        $Z_{i,j} \leftarrow \mu(R_{i,j}, r_i^*)$.    // $Z_{i,j} \leq R_{i,j}$ indicates if the tOCRS selects element $(i,j)$
         when active
        **if** $Z_{i,j} = 1$ **then**
            Allocate item $j$ to agent $i$ with probability $\frac{c-\epsilon}{p_{i,j}^*(r_i^*)}$, where $p_{i,j}^*(r_i^*)$ is the probability that $\mu$
            selects $(i,j)$ conditioned on $i$'s report $r_i^*$ and $R_{i,j}$ being equal to 1.
        **end**
    **end**
**end**

---

*If we only have query access to the tCRS/tOCRS, our mechanisms can be implemented using a* $O(poly(\sum_i |\mathcal{V}_i|, m, \frac{1}{\epsilon}))$ *number of queries in expectation.*

The proof is deferred to Appendix D.

## 3.1 Implementation Considerations

Here, we highlight some implementation details for our framework. First, we give a simple LP that computes optimal ($\alpha = 1$), feasible in expectation, BIC-IR interim allocation and payment rules $(\pi, q)$. Second, we flesh out implementation details of Line 3 of Algorithm 3 and Line 1 of Algorithm 1, flipping a coin with probability $(c - \epsilon)/p_{i,j}^*(v_i)$, when given only black-box access to a tCRS/tOCRS. This is not a straightforward task, since $p_{i,j}^*(v_i)$ might be unknown, and approximating this probability (e.g., via sampling), instead of computing it exactly, results in a violation of the BIC constraint.

**Finding feasible in expectation, optimal interim rules.** The following linear program, (LP1) computes an interim relaxation of the revenue-optimal BIC-IR mechanism.

$$\text{maximize} \quad \sum_{i \in [n]} \sum_{v_i \in \mathcal{V}_i} \Pr[v_i] q_i(v_i)$$

$$\text{s.t.} \quad \sum_{j \in [m]} v_{i,j}\pi_{i,j}(v_i) - q_i(v_i) \geq \sum_{j \in [m]} v_{i,j}\pi_{i,j}(v_i') - q_i(v_i') \quad \forall i \in [n], v_i, v_i' \in \mathcal{V}_i$$

$$\sum_{j \in [m]} v_{i,j}\pi_{i,j}(v_i) - q_i(v_i) \geq 0 \quad \forall i \in [n], v_i \in \mathcal{V}_i$$

$$\pi_i(v_i) \in P_{\mathcal{F}}^i \quad \forall i \in [n], v_i \in \mathcal{V}_i$$

$$\left[\sum_{v_i \in \mathcal{V}_i} \Pr[v_i] \cdot \pi_{i,j}(v_i)\right]_{(i,j) \in [n] \times [m]} \in P_{\mathcal{F}}$$

$$\text{(LP1)}$$

This linear program has $O(m \sum_{i \in [n]} |\mathcal{V}_i|)$ variables and $O(\sum_{i \in [n]} |\mathcal{V}_i|^2)$ constraints, excluding the constraints necessary to represent $P_{\mathcal{F}}^i$ and $P_{\mathcal{F}}$. Therefore, the overall computational complexity of solving this LP depends on whether $P_{\mathcal{F}}$ and $P_{\mathcal{F}}^i$ have an efficient representation.

We note that, in a series of papers, Cai et al. [CDW12a, CDW13a, CDW12b, CDW13b] propose a similar linear program for finding approximately optimal mechanisms. The variables in their LP are the same: the interim allocations and payments. In their framework, however, finding interim allocation rules that can be induced by ex-post feasible allocation rules is crucial. To do so, their constraints, even for simple feasibility sets, are exponential; see Border [Bor07].[2] To solve their LP efficiently they show how to construct (efficient) separation oracles. Once interim allocation and

---
[2]For single item settings, [AFH[+]19] gave a polynomial-sized LP describing Border's inequalities. For general matroids, such a succinct representation is not possible [GNR18].

payment rules are found, they use complicated techniques — techniques that are not amenable to online arrivals of agents — to construct a final mechanism. In contrast, our LP, even though it uses exactly the same variables, only needs to ensure feasibility in expectation. For many settings of interest, our LP has a polynomial-size description, and thus can be solved by any LP solver, making our approach more convenient in practice. Furthermore, our techniques are designed to accommodate for online arrivals of agents.

**Flipping a coin with probability** $p_{i,j}^*(v_i)$**.** To implement Line 3 of Algorithm 3 and Line 1 of Algorithm 1 we can use a Bernoulli factory for division, which, given a $(c - \epsilon)$-coin and a $p_{i,j}^*(v_i)$-coin, outputs a $\frac{c-\epsilon}{p_{i,j}^*(v_i)}$. We know $c - \epsilon$ exactly, so the $(c - \epsilon)$-coin can be implemented trivially. One can implement a $p_{i,j}^*(v_i)$-coin as follows: For each $i' \neq i$ sample $\hat{r}_{i'} \sim \mathcal{D}_{i'}$ and let $R_{i',j} \leftarrow 1$ with probability $b\,\pi_{i',j}(\hat{r}_{i'})$. Also let $R_{i,j} = 1$ with probability $b\,\pi_{i,j}(v_i)$. Querying the tCRS/tOCRS $\mu$ on the active set $R$ returns a set $Z$ of selected elements; output $Z_{i,j}$ as the coin flip for the $p_{i,j}^*(v_i)$-coin.

# 4 Constructing tCRS and tOCRS

In this section, we construct tCRSs and tOCRSs for various feasibility constraints; missing proofs are deferred to Appendix E. First, we prove that for a general family of constraints we call *Vertical-Horizontal (VH) constraints*, it is possible to construct tOCRSs (resp. tCRSs) given OCRSs (resp. CRSs) in a black-box manner.

**Definition 6** (VH Constraints). *We call $\mathcal{F}$ a Vertical-Horizontal (VH) constraint with respect to a ground set $N = \{e_{i,j}\}_{i \in [n], j \in [m]}$ if there exist sets of constraints $\{\mathcal{F}_i\}_{i \in [n]}$, $\{\mathcal{F}^j\}_{j \in [m]}$ such that $I \in \mathcal{F}$ iff (i) $\forall i \in [n]$, $I \cap \{e_{i,j}\}_{j \in [m]} \in \mathcal{F}_i$, (ii) $\forall j \in [m]$, $I \cap \{e_{i,j}\}_{i \in [n]} \in \mathcal{F}^j$.*

**Theorem 2.** *Given $(b, c)$-selectable CRSs (resp. OCRSs) for constraints $\{\mathcal{F}_i\}_{i \in [n]}$ and $(b, c')$-selectable CRSs (resp. OCRSs) for constraints $\{\mathcal{F}^j\}_{j \in [m]}$, there exists a $(b, c \cdot c')$-selectable tCRS (resp. tOCRS) for the induced Vertical-Horizontal constraint $\mathcal{F}$.*

By combining Theorem 2 with known results (e.g., [HKS07, LS18, KS23]) we can get tCRSs and tOCRSs for various settings of interest; we show these applications in Theorem 1 in Section 5.

Next, we construct tOCRSs for knapsack constraints.

**Definition 7** (Knapsack Constraints). *Consider a ground set of elements $N = \{e_{i,j}\}_{i \in [n], j \in [m]}$, where, for each $i, j \in [n] \times [m]$, element $e_{i,j}$ has a weight $k_{i,j}$, and there is a maximum weight $K$. We say that $\mathcal{F}$ is a Knapsack constraint when $I \in \mathcal{F}$ if and only if $I \subseteq N$, and $\sum_{(i,j):e_{i,j} \in I} k_{i,j} \leq K$.*

**Definition 8** (Multi-Choice Knapsack). *Consider a ground set of elements $N = \{e_{i,j}\}_{i \in [n], j \in [m]}$, where, for each $i, j \in [n] \times [m]$, element $e_{i,j}$ has a weight $k_{i,j}$, and there is a maximum weight $K$. We say that $\mathcal{F}$ is a Multi-Choice Knapsack constraint when $I \in \mathcal{F}$ if and only if $I \subseteq N$, $\sum_{(i,j):e_{i,j} \in I} k_{i,j} \leq K$, and, for all $i \in [n]$, $|I \cap \{e_{i,j}\}_{j \in [m]}| \leq 1$.*

The following theorem gives a $(b, \frac{1}{2+8b})$-selectable tOCRS for Knapsack Constraints, for $b \in [0, 1]$. Interestingly, since tOCRSs are OCRSs, our result implies a $(1, 0.1)$-selectable OCRS for knapsack; this is better than the $(1, 0.085)$-selectable OCRS given by Feldman et al. [FSZ21], but not as good as the state-of-the-art $(1, 1/(3 + e^{-2}))$-selectable ($\approx (1, 0.319)$-selectable) OCRS of Jiang et al. [JMZ22].

**Theorem 3.** *For all $b \in [0, 1]$, there exists a $(b, \frac{1}{2+8b})$-selectable tOCRS for Knapsack.*

Next, we give a $(b, \frac{1}{2+7b})$-selectable tOCRS for Multi-Choice constraints, for all $b \in [0, 1]$. The proof of the following theorem is deferred to Appendix E.

**Theorem 4.** *For all $b \in [0, 1]$, there exists a $(b, \frac{1}{2+7b})$-selectable tOCRS for Multi-Choice Knapsack.*

## 4.1 Efficient Implementation Considerations

Theorem 3 and Theorem 4 show that "Knapsack" and "Multi-Choice Knapsack" tOCRSs exist. Both tOCRSs have non-constructive coin-flipping steps (e.g., selecting an active light element with probability $\frac{1}{(1+4b)\Pr[B_{i,j}(d_i)]}$, where $B_{i,j}(d_i)$ is the event that, at the time step when element $e_{i,j}$ is

considered, the total weight of elements in $I$ so far is strictly less than $K/2$, when $d_i$ was sampled from the two-level stochastic process $(\mathcal{D}, bx)$). The following propositions show how to efficiently implement these steps, albeit with a small loss in the performance guarantee.

**Proposition 1.** *We can implement a $\left(b, \frac{1}{2+8b}\left(\frac{1-\delta}{1+10\epsilon}\right)\right)$-selectable tOCRS for Knapsack in time $poly(1/\epsilon^2, \log(1/\delta), m, n)$.*

**Proposition 2.** *We can implement a $\left(b, \frac{1}{2+7b}\left(\frac{1-\delta}{1+8\epsilon}\right)\right)$-selectable tOCRS for Multi-Choice Knapsack in time $poly(1/\epsilon^2, \log(1/\delta), m, n)$.*

## 5 Applications

In this section, we combine results from Sections 3 and 4 to get mechanisms for various problems.

First, our framework can give a sequential mechanism with a $\frac{2e}{e-1}$ ($\approx 3.16$) approximation guarantee for the problem of auctioning off $m$ items to $n$ agents with additive preferences, under "matroid Vertical-Horizontal" constraints $\mathcal{F}$: (i) the set of agents that each item $j$ is allocated to must form a matroid, and (ii) the set of items allocated to each agent $i$ must form a matroid. Observe that, $\mathcal{F}$ is a VH constraint (Definition 6), induced by the aforementioned constraints (i) and (ii). Both (i) and (ii) are matroid constraints, [CVZ14] give a $(1, 1 - \frac{1}{e})$-selectable CRS and [LS18] give a $(1, 1/2)$-selectable OCRS for matroid constraints. Therefore, Theorem 2 (where we use the CRS for the constraints over items and the OCRS for the constraints over agents) implies a $(1, \frac{e-1}{2e})$-selectable tOCRS for $\mathcal{F}$:

**Corollary 1** (Theorem 2 and [LS18]). *There exists a $(1, \frac{e-1}{2e})$-selectable tOCRS for matroid Vertical-Horizontal constraints.*

Corollary 1 and Theorem 1 readily give the following result.

**Application 1** (Corollary 1 and Theorem 1). *Consider the problem of auctioning off $m$ items to $n$ agents with additive preferences, such that the set of agents that each item $j$ is allocated to must form a matroid, and the items allocated to each agent must form a matroid. There exists a sequential, BIC-IR mechanism that guarantees a $\frac{2e}{e-1}$ ($\approx 3.16$) approximation to the optimal revenue.*

Notably, the previously best-known approximation guarantee for a sequential mechanism for even a special case of this problem (agents with preferences that are constrained additive with a matroid constraint, and each item can be allocated to at most one agent) was 70 [CZ17].

CRSs and OCRSs with approximation factors better than 2 (i.e. better than the result of [LS18] for general matroids) are possible for some special cases. E.g., for $k$-uniform matroids, [HKS07] give a $\left(1 - O\left(\sqrt{\frac{\log k}{k}}\right)\right)$-selectable OCRS, and [KS23] give a $\left(1, 1 - \binom{n}{k}\left(1 - \frac{k}{n}\right)^{n+1-k}\left(\frac{k}{n}\right)^k\right)$-selectable CRS, where $n$ is the number of elements (for a fixed $k$, this approaches $\left(1, 1 - e^{-k}\frac{k^k}{k!}\right)$). Combining with Theorem 2 we can get tOCRSs/tCRSs for these cases, and applying Theorem 1 gives an overall (sequential for tOCRSs) mechanism with a slightly improved guarantee.

We note that, depending on the choice of matroids, (LP1) might not be efficiently computable. For example, the representation of $P_{\mathcal{F}}$ might require an exponential (in $n$, $m$ and $\sum_{i\in[n]}|\mathcal{V}_i|$) number of inequalities; in such cases, our approach does not give an end-to-end efficient procedure for finding a mechanism. However, if one is given feasible in expectation, BIC-IR and approximately optimal interim rules, all remaining steps of our framework can be efficiently computed.

Finally, we combine Theorem 1 with our tOCRSs for Knapsack and Multi-Choice Knapsack.

**Application 2** (Theorem 3, Theorem 4, and Theorem 1). *Consider the problem of auctioning off $m$ items to $n$ agents with additive preferences, where each item $j \in [m]$ has some weight $k_j$ and the total weight of items sold cannot exceed $K$. There exists an efficiently computable, sequential, BIC-IR mechanism that guarantees a $10$ approximation to the optimal revenue. Additionally, if each agent $i$ can get at most one item, there exists an efficiently computable, sequential, BIC-IR mechanism that guarantees a $9$ approximation to the optimal revenue.*

Recently, [ABM+22] gave a simple and approximately optimal, with respect to welfare, mechanism for the Rich-Ads problem. Application 2 readily gives a sequential, approximately optimal, with respect to revenue, mechanism for the same problem.

Finally, by creating a meta-item for each possible subset of items (where the weight of the meta-item is simply the sum of weights from the corresponding subset), our approach gives an approximately optimal, sequential, BIC-IR mechanism for *arbitrary* valuation functions (subject to a knapsack constraint). For a logarithmic (in $n$) number of items, this approach gives a computationally efficient procedure as well (but, of course, this is not true in general).

**Application 3** (Theorem 4 and Theorem 1)**.** *Consider the problem of auctioning off $m$ items to $n$ agents with arbitrary valuation functions, where each item $j \in [m]$ has some weight $k_j$ and the total weight of items sold cannot exceed $K$. There exists a sequential, BIC-IR mechanism that guarantees a $9$ approximation to the optimal revenue.*

## Acknowledgments

Marios Mertzanidis and Alexandros Psomas are supported in part by an NSF CAREER award CCF-2144208, and a research award from the Herbert Simon Family Foundation.

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

# A  Further Related Work

**Approximations in Bayesian Mechanism Design.**    There is a rich body of literature on approximately optimal mechanism design, as discussed in the introduction. The work most related to ours is [Ala14] which provides a framework for designing multi-agent mechanisms by a reduction to single agent mechanisms via an ex-ante relaxation and using an online rounding procedure or OCRS. This framework yielded approximately optimal non-sequential and sequential mechanisms for several fundamental settings including selling $k$ identical items to $n$ additive buyers, who may be subject to additional constraints such as budget constraints, although does not support any inter-buyer constraints (except for per item supply constraints). In particular, a sequential posted price mechanism can obtain a $1 - O(1/\sqrt{k})$ to the optimal revenue. In the special case of unit-demand buyers with independent values for heterogeneous items (where each item can be allocated to at most one bidder), [Ala14] establishes a sequential posted price mechanism that obtains a $4$-approximation—recovering a prior result of [CHMS10]. Sequential mechanisms have been of particular interest due to their simplicity in implementation. The [Ala14] framework (along with other influential techniques like [CDW19] duality) has enabled the design of approximately optimal and simple sequential mechanisms in various follow-up works. [AFHH13] show that revenue maximization in a multi-dimensional multi-agent setting reduces (exactly or approximately) to a single-agent problem in a variety of settings, including risk-averse agents. [AW18] use random order OCRS to design a $(k + 4)$-approximation mechanism for unit demand bidders under $k$-matroid feasibility constraints. [CM16] consider a more general case where agents have constrained additive valuations (aka matroid rank valuations) and provide a simple sequential two-part tariff mechanism that obtains a constant approximation. This constant was then improved to $70$ by [CZ17], who also extend their results to the setting where agents have XOS (and subadditive) valuations over $m$ heterogeneous, independent items, and [DKL20] provide improved approximation ratio of $O(\log \log m)$ for subadditive valuations. Crucially, these works assume that the agents' valuations are independent over the item. For the setting of correlated items and constrained additive buyers (with matroid constraints), [Ala14] provides a sequential mechanism that is a $2$-approximation to the optimal revenue. However, this mechanism is not "simple" (unlike posted price or two-tariff). Indeed when the item distributions are correlated no "simple" mechanism can obtain any non-trivial approximation [HN19, BCKW15]. A recent line of work circumvents these impossibilities by studying approximation guarantees of simple mechanisms with respect to a weaker benchmark called buy-many mechanisms [BCKW15, CRTT23, CCD$^+$24].

**CRSs, OCRSs, and their applications.**    Contention resolution schemes were defined by [CVZ14], and extended by [FSZ21] to online settings. CRSs and OCRSs have since found numerous applications in Bayesian and stochastic online optimization, such as stochastic probing [FTW$^+$21, GN13] (with applications to posted price mechanisms), prophet inequalities [EFGT20] (in fact, OCRSs are equivalent to ex-ante prophet inequalities [LS18]), pricing problems [PRSW22, CCD$^+$24], and network revenue management [MMZ24]. In this paper, we define a type of dependent CRSs and OCRSs, which we call two-level CRSs/OCRSs. Online dependent rounding schemes, under the name of ODRS, were introduced by [NSW23] in the context of rounding fractional bipartite $b$-matchings online; here we give a different name to our schemes to highlight the specific dependence we can handle. There has also been a recent work studying OCRSs with limited correlation or pairwise independence [GHKL24, DKP23]. A special case of tOCRS called random-element OCRS, where at most one element from $\{e_{ij}\}_j$ becomes active for each $i$, is studied in [MMZ24] for constraints induced by "$L$-bounded products". Other works study OCRSs under negative correlations [Dug19, QS22].

# B  Additional Preliminaries

## B.1  CRS and tCRS Preliminaries

**Definition 9** (Contention Resolution Scheme (CRS) [CVZ14])**.** *Let $b, c \in [0, 1]$. For every $x \in P_{\mathcal{F}}$, let $R(b\,x)$ be a random subset of active elements, where element $i \in N$ is active with probability $b\,x_i$, independently. A $(b, c)$-selectable Contention Resolution scheme (CRS) $\mu$ for $P_{\mathcal{F}}$ is a (possibly randomized) procedure that, given a set of active elements $R(b\,x)$ returns a set $\mu(R(b\,x)) = I \subseteq R(b\,x)$, such that (i) $I \in \mathcal{F}$, and (ii) $\Pr[i \in I | i \in R(b\,x)] \geq c, \forall i \in N$.*

**Definition 10** (Two-level CRS (tCRS))**.** *Let $b, c \in [0, 1]$. Let $(\mathcal{D}, x)$ be a two-level stochastic process that is feasible with respect to $\mathcal{F}$, and let $R(\mathcal{D}, b\,x) \subseteq N$ be a random set of elements obtained by*

sampling from $(\mathcal{D}, b\,x)$. A $(b, c)$-selectable two-level CRS (tCRS) $\mu$ for $\mathcal{F}$ is a (possibly randomized) procedure that, given a set of active elements $R(\mathcal{D}, b\,x) \subseteq N$ and realizations $d = (d_1, \ldots, d_n)$ that were sampled in the first level of $(\mathcal{D}, x)$, returns a set $\mu(R(\mathcal{D}, b\,x), d) = I \subseteq R(\mathcal{D}, b\,x)$, such that (i) $I \in \mathcal{F}$, and (ii) $\Pr[i \in I | i \in R(\mathcal{D}, b\,x)] \geq c, \forall i \in N$.

## B.2  Bernoulli Factories

Bernoulli factories were introduced by Keane and O'Brien [KO94], where they are defined as follows.

**Definition 11** (Bernoulli Factory). *Given a function $f : (0, 1) \to (0, 1)$, a Bernoulli factory for $f$ outputs a sample of a Bernoulli variable with bias $f(p)$ (i.e. an $f(p)$-coin), given black-box access to independent samples of a Bernoulli distribution with bias $p \in (0, 1)$ (i.e. a $p$-coin).*

As an illustrative example, imagine that we are given a $p$-coin, a coin that outputs 1 with probability $p$ and 0 otherwise. Our goal is to create a new coin that outputs 1 with probability $f(p) = p^2$. The complication here is that we do not know the value of $p$. $f(p) = p^2$ can be implemented as follows: flip the $p$-coin twice. If both are 1 then output 1 (otherwise output 0). We include additional examples of Bernoulli factories in Appendix B.2. Bernoulli factories have recently been used in mechanism design in the context of black-box reductions [DHKN21, COVZ21]. In this paper, we make use of a Bernoulli factory for division: given one $p_0$-coin and one $p_1$-coin, implement $f(p_0, p_1) = p_0/p_1$ for $p_1 - p_0 \geq \delta$. This problem was considered by [NP05] but their construction is rather involved. Instead, consider Algorithm 2, the Bernoulli Division factory of [Mor21].

---

**ALGORITHM 2:** Bernoulli Division [Mor21]

**while** *true* **do**
    $X \sim Bern[1/2]$.
    **if** *X = 0* **then**
        $W \sim Bern[p_0]$.
        **if** $W = 1$ **then**
           |  return 1
        **end**
    **end**
    **else**
        $W \sim Bern[p_1 - p_0]$.
        **if** $W = 1$ **then**
           |  return 0
        **end**
    **end**
**end**

---

**Lemma 1** ([Mor21]). *Given a $p_0$-coin and a $p_1$-coin, assume $p_1 - p_0 \geq \delta$, and let $N$ be the number of tosses required. Then, Algorithm 2 is a Bernoulli factory for $(p_0/p_1)$ which satisfies $\mathbb{E}[N] \leq \frac{22.12}{p_1}(1 + \delta^{-1})$.*

Although the process and its correctness are fully described by [Mor21], the end-to-end expected number of tosses is not explicitly calculated. For completeness, we show these calculations, and also argue that Algorithm 2 is a Bernoulli factory for $(p_0/p_1)$.

*Proof of Lemma 1.* Consider the following distribution:

$$\Pr[Y_k = y_k] = \begin{cases} 1 - \frac{1}{2}p_1 & \text{if } y_k = -1 \\ \frac{1}{2}(p_1 - p_0) & \text{if } y_k = 0 \\ \frac{1}{2}p_0 & \text{if } y_k = 1 \end{cases}$$

Then it is not difficult to see that:

$$\sum_{k=0}^{\infty} \Pr[Y_k = 1] \prod_{i=0}^{k-1} \Pr[Y_i = -1] = \frac{1}{2}p_0 \sum_{k=0}^{\infty} \left(1 - \frac{1}{2}p_1\right)^k = \frac{p_0}{p_1},$$

and thus Algorithm 2 is a valid Bernoulli factory for $p_0/p_1$.

Let $N_t$ be the random variable that represents the number of tosses at round $t$, and let $X_t$ be a random variable that is 1 if the experiment lasts at least $t$ rounds and 0 otherwise. Then $N = \sum_{t=1}^{\infty} N_t X_t$. Linearity of expectation implies that $\mathbb{E}[N] = \sum_{t=1}^{\infty} \mathbb{E}[N_t X_t]$, however $X_t$ and $N_t$ are independent and thus $\mathbb{E}[N] = \sum_{t=1}^{\infty} \mathbb{E}[N_t]\mathbb{E}[X_t]$. From [Mor21] (Proposition 2.27) we know that $\mathbb{E}[N_t] \leq 11.06(1 + \delta^{-1})$. On the other hand $\sum_{t=1}^{\infty} \mathbb{E}[X_t] = \sum_{t=1}^{\infty} \left(1 - \frac{1}{2}p_i\right)^t = \frac{2}{p_1}$. Combining the above concludes the proof. $\qquad\square$

Finally, we outline a few Bernoulli factories and their construction to help the reader gain some intuition:

1. Bernoulli Negation: Given a $p$-coin, implement $f(p) = 1 - p$. This can be implemented with one sample from the $p$-coin:
   - $P \sim Bern[p]$.
   - If $P = 0$ output 1 (otherwise output 0).

2. Bernoulli Down Scaling: Given a $p$-coin, implement $f(p) = \lambda \cdot p$ for a constant $\lambda \in [0,1]$. This can be implemented with one sample from the $p$-coin:
   - Draw $\Lambda \sim Bern[\lambda]$ and $P \sim Bern[p]$.
   - Output $\Lambda \cdot P$.

3. Bernoulli Averaging: Given one $p_0$-coin and one $p_1$-coin, implement $f(p_0, p_1) = \frac{p_0 + p_1}{2}$. This can be implemented with one sample from the $p_0$-coin or one sample from the $p_1$-coin:
   - Draw $Z \sim Bern[1/2]$, $P_0 \sim Bern[p_0]$, and $P_1 \sim Bern[p_1]$.
   - Output $P_Z$.

4. Bernoulli Doubling: Given a $p$-coin, implement $f(p) = 2p$ for $p \in (0, 1/2 - \delta]$. This can be implemented with $O(1/\delta)$ samples in expectation from the $p$-coin. The algorithm was introduced by [NP05].

5. Bernoulli Addition: Given one $p_0$-coin and one $p_1$-coin, implement $f(p_0, p_1) = p_0 + p_1$ for $p_0 + p_1 \in [0, 1 - \delta]$. This can be implemented with $O(1/\delta)$ samples in expectation from the $p_0$-coin and $p_1$-coin:
   - Use Bernoulli Averaging to create a $\frac{p_0 + p_1}{2}$-coin.
   - Use Bernoulli Doubling on the $\frac{p_0 + p_1}{2}$-coin.

6. Bernoulli Subtraction [Mor21]: Given one $p_0$-coin and one $p_1$-coin, implement $f(p_0, p_1) = p_1 - p_0$ for $p_1 - p_0 \geq \delta$. This can be implemented with $O(1/\delta)$ samples in expectation from the $p_0$-coin and $p_1$-coin:
   - Use Bernoulli Negation on the $p_1$-coin to create a $1 - p_1$-coin.
   - Use Bernoulli Addition on the $1 - p_1$-coin and $p_0$-coin to create a $1 - p_1 + p_0$-coin.
   - Use Bernoulli Negation on the $1 - p_1 + p_0$-coin.

## C  Mechanisms from two-level CRSs

The input to the tCRS framework is (i) a BIC-IR interim allocation, payment rule pair $(\pi, q)$ that is feasible in expectation and is an $\alpha$ approximation to the optimal revenue, (ii) a $(b, c)$-selectable tCRS for $\mathcal{F}$, and (iii) a parameter $\epsilon \geq 0$. Our frameworks produce BIC-IR, $\frac{\alpha}{b(c-\epsilon)}$ approximately optimal mechanisms for agents with constrained (with respect to $\mathcal{F}$) additive valuations.

Given a tCRS, our framework, Algorithm 3, works as follows. First, each agent $i$ reports $r_i^* \in \mathcal{V}_i$ and pays $b(c - \epsilon) \cdot q_i(r_i^*)$. We then construct a set $R = \{x_{i,j}\}_{i\in[n], j\in[m]}$ of $n \cdot m$ elements, one for every agent, item pair, where $x_{i,j} = 1$ (i.e., element $(i,j)$ is active) with probability $b \cdot \pi_{i,j}(r_i^*)$. We query the tCRS on input $R$, to get back a subset $Z$ of selected elements (which is in $\mathcal{F}$, by definition). We flip an additional coin to decide whether to keep an element $(i,j)$, i.e. whether we should allocate item $j$ to agent $i$. Recall that this last coin essentially balances the randomness of the tCRS and ensures that the probability that agent $i$ gets item $j$ when they report $r_i^*$ is exactly $b(c - \epsilon)\pi_{i,j}(r_i^*)$, which, combined with the chosen payment, ensures that BIC and BIR are satisfied.

---
**ALGORITHM 3:** Our framework for tCRSs
---
**Input:** allocation, payment rule pair $(\pi, q)$, $(b, c)$-selectable tCRS $\mu$, parameter $\epsilon \geq 0$.

---
**for** *each* $i \in [n]$ **do**
  | Agent $i$ reports $r_i^* \in \mathcal{V}_i$ and pays $b\,(c - \epsilon)\,q_i(r_i^*)$.
**end**
Construct $R = \{x_{i,j}\}_{i \in [n], j \in [m]}$, where $x_{i,j} \leftarrow 1$ with probability $b\,\pi_{i,j}(r_i^*)$, and $x_{i,j} \leftarrow 0$ otherwise.
$Z = \{Z_{i,j}\}_{i \in [n], j \in [m]} \leftarrow \mu(R, r^*)$. `// Z ⊆ R is the set of elements that the tCRS picks`
**for** *each element* $(i, j) \in [n] \times [m]$ *with* $Z_{i,j} = 1$ **do**
  | Allocate item $j$ to agent $i$ with probability $\frac{c - \epsilon}{p_{i,j}^*(r_i^*)}$, where $p_{i,j}^*(r_i^*)$ is the probability that $\mu$ selects $(i, j)$
  | conditioned on $i$'s report $r_i^*$ and $x_{i,j}$ being equal to 1.
**end**

---

# D    Missing Proofs from Section 3

*Proof of Theorem 1.* First, we argue that our frameworks output allocations in $\mathcal{F}$. By definition, and assuming truthful reports, the interim allocation $\pi$ defines a feasible (for $\mathcal{F}$) two-level stochastic process, where we first sample $v_i$s independently, and then $\pi_{i,j}(v_i)$. Let $x \in R(b\,\pi)$ be the input to the $(b, c)$-selectable tCRS (resp. for tOCRS); by definition, the set of selected elements $Z$ satisfies $Z \in \mathcal{F}$. We allocate a subset of $Z$, thus our allocations are feasible, since $\mathcal{F}$ is downward closed. Furthermore, for any element $z_{i,j}$, $\Pr[z_{i,j} = 1 | x_{i,j} = 1] = p_{i,j}^*(v_i) \geq c$, and thus $\frac{c - \epsilon}{p_{i,j}^*(v_i)}$ is a probability. Second, we argue that our mechanisms are BIC. From the perspective of agent $i$, a report $r_i \in \mathcal{V}_i$ costs $b\,(c - \epsilon) \cdot q_i(r_i)$ and allocates item $j$ with probability $b\,\pi_{i,j}(r_i) \cdot p_{i,j}^*(r_i) \cdot \frac{c - \epsilon}{p_{i,j}^*(r_i)} = b\,(c - \epsilon) \cdot \pi_{i,j}(r_i)$, and therefore translates to an expected utility of costs $b\,(c - \epsilon) \sum_{j \in [m]} v_{i,j}\pi_{i,j}(r_i) - b\,(c - \epsilon) \cdot q_i(r_i)$; since $(\pi, q)$ is BIC, so is the mechanism we output. Near-identical arguments imply the BIR guarantee and revenue guarantees.

When given only black-box access to a tCRS/tOCRS, it is not immediately clear how one can flip a coin with probability exactly $(c - \epsilon)/p_{i,j}^*(v_i)$ (efficiently or otherwise), as needed in Line 3 of Algorithm 3 and Line 1 of Algorithm 1. Using a Bernoulli factory for division (such as the result of [Mor21] discussed in Section 2), this step can be implemented using $\frac{22.12}{p_{i,j}^*(v_i)}(1 + \epsilon^{-1}) \in O(1/\epsilon)$ calls in expectation (Lemma 1), assuming that $c \leq p_{i,j}^*(v_i)$ is a constant. Overall, we have $O(poly(\sum_i |\mathcal{V}_i|, m, \frac{1}{\epsilon}))$ queries in expectation for the entire execution of a framework; we discuss implementation details in Section 3.1. $\qquad\square$

# E    Missing Proofs from Section 4

*Proof of Theorem 2.* Let $(\mathcal{D}, b\,x)$ be the two-level stochastic process through which elements become active. Let $\mu_i$ be the $(b, c)$-selectable CRS/OCRS for constraint $\mathcal{F}_i$, for $i \in [n]$, and let $\mu^j$ be the $(b, c')$-selectable CRS/OCRS for constraint $\mathcal{F}^j$, for $j \in [m]$. Given a set of active elements $R(\mathcal{D}, b\,x)$ sampled from $(\mathcal{D}, b\,x)$, our tCRS/tOCRS selects element $e_{i,j} \in R(\mathcal{D}, b\,x)$ if (i) $\mu^j$ on input $R(\mathcal{D}, b\,x) \cap \{e_{i,j}\}_{i \in [n]}$ selects $e_{i,j}$, and (ii) $\mu_i$ on input $R(\mathcal{D}, b\,x) \cap \{e_{i,j}\}_{j \in [m]}$ selects $e_{i,j}$ (noting that this is process does make decisions online when $\mu^j$ and $\mu_i$ are OCRSs and are queried in an online fashion).

Let $A_{i,j}$ be the event that $e_{i,j} \in R(\mathcal{D}, b\,x)$. By the definition of a two-level stochastic process, event $A_{i,j}$ is independent from event $A_{i',j}$, for all $j \in [m]$ and $i' \in [n]$ such that $i \neq i'$. Therefore, the CRSs/OCRSs $\mu^j$, $j \in [m]$, are queried about elements that become active independently (as required by the definition of a CRS/OCRS). Now, overloading notation, let $A_{i,j}(d_i)$ be the event that $e_{i,j} \in R(\mathcal{D}, b\,x)$ given that $d_i$ was sampled from $\mathcal{D}_i$. By the definition of a two-level stochastic process, event $A_{i,j}(d_i)$ is independent from event $A_{i,j'}(d_i)$, for all $j \neq j' \in [m]$ and all $i \in [n]$. Therefore, the CRSs/OCRSs $\mu_i$, $i \in [n]$, are queried about elements that become active independently (as required by the definition of a CRS/OCRS). Let $B_{i,j}$ be the event that $\mu^j$ selects an active element $e_{i,j} \in R(\mathcal{D}, b\,x)$ on input $R(\mathcal{D}, b\,x) \cap \{e_{i,j}\}_{i \in [n]}$, and note that, since $\mu^j$ is a $(b, c')$-selectable CRS/OCRS, we have that then $\Pr[B_{i,j} | A_{i,j}] \geq c'$. Similarly, let $C_{i,j}$ be the event that $\mu_i$ selects an active element $e_{i,j} \in R(\mathcal{D}, b\,x)$ on input $R(\mathcal{D}, b\,x) \cap \{e_{i,j}\}_{j \in [m]}$; $\Pr[C_{i,j} | A_{i,j}] \geq c$. Finally,

conditioned on $A_{i,j}$ events $B_{i,j}$ and $C_{i,j}$ are conditionally independent due to the definition of a two-level stochastic process. Thus, $\Pr[B_{i,j} \cap C_{i,j}|A_{i,j}] \geq c\,c'$, which concludes the proof. $\qquad \square$

*Proof of Theorem 3.* Let $(\mathcal{D}, bx)$ be the two-level stochastic process through which elements become active. Let $k_{i,j}$ be the weight of element $e_{i,j}$, and $K$ bet the total weight. Let $S_h = \{e_{i,j} : i \in [n], j \in [m]|k_{i,j} > K/2\}$ be the set of elements whose weight is at least $K/2$, the "heavy elements," and let $S_\ell = \{e_{i,j} : i \in [n], j \in [m]|k_{i,j} \leq K/2\}$ be the set of "light elements." Our tOCRS is randomized: with probability $1/2$ we run a scheme that considers only heavy elements, the "heavy scheme," and with probability $1/2$ we run a scheme that considers only light elements, the "light scheme." In both cases, we use $I$ to indicate the set of elements we output. Without loss of generality (from the definition of a tOCRS) we assume that elements arrive in the order $e_{1,1}, e_{1,2}, \ldots, e_{n,m}$.

For the case of the heavy scheme, it is obvious that we can only take one heavy element. We initialize $I = \emptyset$ and consider elements sequentially. Let $A_{i,j}(d_i)$ be the event that $I$ is empty until element $e_{i,j}$ is considered when we run the heavy scheme and $d_i$ was sampled from the two-level stochastic process $(\mathcal{D}, bx)$. If element $e_{i,j}$ is active, $e_{i,j} \in S_h$ and $I = \emptyset$, then with probability $\frac{1}{(1+4b)\Pr[A_{i,j}(d_i)]}$ we set $I \leftarrow \{e_{i,j}\}$, otherwise we move on to the next element. Assuming that $\frac{1}{(1+4b)\Pr[A_{i,j}(d_i)]}$ is a valid probability, each heavy element is selected with probability exactly $\Pr[A_{i,j}(d_i)]\frac{1}{(1+4b)\Pr[A_{i,j}(d_i)]} = \frac{1}{(1+4b)}$ when we run the heavy scheme. Towards proving that $\Pr[A_{i,j}(d_i)] \geq \frac{1}{(1+4b)}$ we have

$$
\Pr[A_{i,j+1}(d_i)] = 1 - \left( \sum_{i' \leq i} \sum_{j':e_{i',j'} \in S_h} \Pr[e_{i',j'} \text{ picked}] + \sum_{j' < j+1:e_{i,j'} \in S_h} \Pr[e_{i,j'} \text{ picked}] \right)
$$

$$
= 1 - \left( \sum_{i' \leq i} \sum_{j':e_{i',j'} \in S_h} \Pr[e_{i',j'} \text{ active}] \cdot \Pr[e_{i',j'} \text{ picked}|e_{i',j'} \text{ active}] \right.
$$
$$
\left. + \sum_{j' < j+1:e_{i,j'} \in S_h} \Pr[e_{i,j'} \text{ active}] \Pr[e_{i,j'} \text{ picked}|e_{i,j'} \text{ active}] \right)
$$

$$
= 1 - \frac{1}{1+4b} \left( \sum_{i' \leq i} \sum_{j':e_{i',j'} \in S_h} \Pr[e_{i',j'} \text{ active}] + \sum_{j' < j+1:e_{i,j'} \in S_h} \Pr[e_{i,j'} \text{ active}] \right)
$$

$$
= 1 - \frac{1}{1+4b} \left( \sum_{i' \leq i} \sum_{j':e_{i',j'} \in S_h} w_{i',j'} + \sum_{j' < j+1:e_{i,j'} \in S_h} x_{i,j'}(d_i) \right)
$$

$$
> 1 - \frac{4b}{1+4b} = \frac{1}{1+4b},
$$

where $w_{i,j} = \sum_{d_i \in \mathcal{V}_i} \Pr[\mathcal{D}_i = d_i] \, x_{i,j}(d_i)$, is the probability that $e_{i,j}$ is active in $(\mathcal{D}, bx)$.

For the case of the light scheme, we again initialize $I = \emptyset$. Let $B_{i,j}(d_i)$ be the event that, at the time step when element $e_{i,j}$ is considered, the total weight of elements in $I$ so far is strictly less than $K/2$, when $d_i$ was sampled from the two-level stochastic process $(\mathcal{D}, bx)$. We consider each (light) element $e_{i,j}$ one at a time, and if $e_{i,j}$ is active and the weight of elements in $I$ is less than $K/2$, we set $I \leftarrow I \cup \{e_{i,j}\}$ with probability $\frac{1}{(1+4b)\Pr[B_{i,j}(d_i)]}$; otherwise we move on to the next element. Again, if $\frac{1}{(1+4b)\Pr[B_{i,j}(d_i)]}$ is a valid probability, each light element is selected with probability exactly $\Pr[B_{i,j}(d_i)]\frac{1}{(1+4b)\Pr[B_{i,j}(d_i)]} = \frac{1}{(1+4b)}$ when we run the light scheme. It therefore remains to prove that $\Pr[B_{i,j}(d_i)] \geq \frac{1}{(1+4b)}$. Let $W_{i,j}(d_i)$ be the random variable that represents the total weight of elements in $I$ at the time when we consider elements $e_{i,j}$ when $d_i$ was the sample from $(\mathcal{D}, bx)$. We have that:

$$
\mathbb{E}\left[W_{i,j}(d_i)\right] = \underbrace{\sum_{i' < i} \sum_{d_{i'} \in \mathcal{V}_{i'}} \Pr[\mathcal{D}_{i'} = d_{i'}] \sum_{j':e_{i',j'} \in S_\ell} \frac{1}{(1+4b)\Pr[B_{i',j'}(d_{i'})]} \Pr[B_{i',j'}(d_{i'})]bx_{i',j'}(d_{i'})k_{i',j'}}_{\text{Contribution from agents before } i}
$$

$$+ \sum_{\substack{j' < j: \\ e_{i,j'} \in S_\ell}} \frac{1}{(1 + 4b) \Pr[B_{i,j'}(d_i)]} \underbrace{\Pr[B_{i,j'}(d_i)] b x_{i,j'}(d_i) k_{i,j'}}$$

Contribution from $i$'s items before $j$

$$= \frac{b}{1 + 4b} \left( \sum_{i' < i} \sum_{j' : e_{i',j'} \in S_l} w_{i',j'} k_{i',j'} + \sum_{\substack{j' < j: \\ e_{i,j'} \in S_l}} x_{i,j'}(d_i) k_{i,j'} \right)$$

$$\leq \frac{2b}{1 + 4b} K. \qquad \text{(Feasibility of } (\mathcal{D}, bx))$$

Therefore, we have

$$
\begin{aligned}
\Pr[B_{i,j}(d_i)] &= \Pr[W_{i,j}(d_i) < K/2] \\
&= 1 - \Pr[W_{i,j}(d_i) \geq K/2] \\
&\geq 1 - \frac{\frac{2b}{1+4b} K}{K/2} \qquad \text{(Markov's Inequality)} \\
&= \frac{1}{1 + 4b}.
\end{aligned}
$$

Since we run each scheme with probability $1/2$, for an element $e_{i,j} \in S_h$ that is active we have:

$$\Pr[e_{i,j} \in I] = \Pr[\text{"heavy scheme"}] \Pr[e_{i,j} \in I | \text{"heavy scheme"}] \geq \frac{1}{2} \frac{1}{1 + 4b} = \frac{1}{2 + 8b}.$$

Similarly, for an active element $e_{i,j} \in S_\ell$, $\Pr[e_{i,j} \in I] \geq 1/(2 + 8b)$, concluding the proof. $\qquad \square$

*Proof of Theorem 4.* Let $(\mathcal{D}, bx)$ be the two-level stochastic process through which elements become active. Let $k_{i,j}$ be the weight of element $e_{i,j}$, and $K$ bet the total weight. Our overall approach is similar to Theorem 3.

Let $S_h = \{e_{i,j} : i \in [n], j \in [m] | k_{i,j} > K/2\}$ be the set of "heavy elements" and $S_\ell = \{e_{i,j} : i \in [n], j \in [m] | k_{i,j} \leq K/2\}$ be the set of "light elements." Our tOCRS is randomized: with probability $\frac{1+4b}{2+7b}$ we run a scheme that considers only heavy elements, the "heavy scheme," and with probability $\frac{1+3b}{2+7b}$ we run a scheme that considers only light elements, the "light scheme." In both cases, we use $I$ to indicate the set of elements we output. Without loss of generality (from the definition of a tOCRS) we assume that elements arrive in the order $e_{1,1}, e_{1,2}, \ldots, e_{n,m}$.

For the heavy scheme, initialize $I = \emptyset$ and consider elements sequentially. Let $A_{i,j}(d_i)$ be the event that $I = \emptyset$ when element $e_{i,j}$ is considered when we run the heavy scheme and $d_i$ was sampled from the two-level stochastic process $(\mathcal{D}, bx)$. If element $e_{i,j}$ is active, heavy, and $I = \emptyset$, then with probability $\frac{1}{(1+4b) \Pr[A_{i,j}(d_i)]}$ we set $I \leftarrow \{e_{i,j}\}$; otherwise we move on to the next element. The analysis in Theorem 3 proves that $\frac{1}{(1+4b) \Pr[A_{i,j}(d_i)]}$ is a valid probability. Notice that each heavy element is selected with probability exactly $\Pr[A_{i,j}(d_i)] \frac{1}{(1+4b) \Pr[A_{i,j}(d_i)]} = \frac{1}{(1+4b)}$ given that we run the heavy scheme.

Now consider the light case. We initialize $I = \emptyset$. Let $B_i$ be the event that $\sum_{i',j : e_{i',j} \in I, i' < i} k_{i',j} < K/2$. Also let $C_{i,j}(d_i)$ be the event that, when $d_i$ was sampled from the two-level stochastic process $(\mathcal{D}, bx)$, $\forall j' \in S_\ell, j' < j : e_{i,j'} \notin I$. We consider each light element $e_{i,j} \in S_\ell$ sequentially and, if it is active, and the weight of elements in $I$ is less than $K/2$, and no other element of $i$ has been selected, we set $I \leftarrow I \cup \{e_{i,j}\}$ with probability $\frac{1}{(1+3b) \Pr[B_i \cap C_{i,j}(d_i)]}$; otherwise we move on to the next element. If $\Pr[B_i \cap C_{i,j}(d_i)] \geq \frac{1}{(1+3b)}$ (i.e., if $\frac{1}{(1+3b) \Pr[B_i \cap C_{i,j}(d_i)]}$ is a valid probability), each light element is selected with probability exactly $\frac{1}{1+3b}$ when we run the light scheme. Towards proving that $\Pr[B_i \cap C_{i,j}(d_i)] \geq \frac{1}{(1+3b)}$, let $W_i$ elements in $I$ when we consider the first element of $i$. We have:

$$\mathbb{E}[W_i] = \sum_{i'<i} \sum_{d_{i'} \in \mathcal{V}_{i'}} \sum_{j \in S_\ell} \Pr[\mathcal{D}_{i'} = d_{i'}] \Pr[B_{i'} \cap C_{i',j}(d_{i'})] \frac{b}{(3b+1)\Pr[B_{i'} \cap C_{i',j}(d_{i'})]} bx_{i',j}(d_{i'})k_{i',j}$$

$$= \frac{b}{1+3b} \sum_{i'<i} \sum_{j \in S_\ell} w_{i',j}k_{i',j}$$

$$\leq \frac{b}{1+3b}K. \qquad \text{(Knapsack feasibility of } (\mathcal{D}, bx))$$

Therefore:

$$\Pr[B_i] = \Pr[W_i < K/2] = 1 - \Pr[W_i \geq K/2] \geq^{\text{(Markov's Inequality)}} 1 - \frac{\frac{b}{1+3b}K}{K/2} = \frac{1+b}{1+3b}.$$

On the other hand,

$$\Pr[C_{i,j+1}(d_i)|B_i] = \Pr[C_{i,j}(d_i)|B_i]\left(1 - \frac{1}{(1+3b)\Pr[B_i \cap C_{i,j}(d_i)]}bx_{i,j}(d_i)\right)$$

$$= \Pr[C_{i,j}(d_i)|B_i] - \frac{1}{(1+3b)\Pr[B_i]}bx_{i,j}(d_i)$$

$$= 1 - \frac{1}{(1+3b)\Pr[B_i]} \sum_{j' \leq j : j' \in S_\ell} bx_{i,j}(d_i)$$

$$\geq 1 - \frac{b}{(1+3b)\Pr[B_i]}. \qquad \text{(multi-choice feasibility of } (\mathcal{D}, bx))$$

Combining the above we have that:

$$\Pr[B_i \cap C_{i,j}(d_i)] = \Pr[B_i]\Pr[C_{i,j+1}(d_i)|B_i] \geq \Pr[B_i] - \frac{b}{(1+3b)} = \frac{1}{1+3b}.$$

We run the heavy scheme with probability $\frac{1+4b}{2+7b}$; thus, for an active element $e_{i,j} \in S_h$ we have:

$$\Pr[e_{i,j} \in I] = \Pr[\text{"heavy scheme"}]\Pr[e_{i,j} \in I|\text{"heavy scheme"}] \geq \frac{1+4b}{2+7b}\frac{1}{1+4b} = \frac{1}{2+7b}.$$

We can similarly show that active light elements are also selected with probability at least $\frac{1}{2+7b}$. This concludes the proof of Theorem 4. $\qquad \square$

*Proof of Proposition 1.* In the proof of Theorem 3 we give an exact formula for the probability that $A_{i,j}(d_i)$ occurs; therefore, the only step we cannot directly implement from the procedure outlined in the proof of Theorem 3 is the toss of the $\frac{1}{(1+4b)\Pr[B_{i,j}(d_i)]}$ coin. Even when given a $\Pr[B_{i,j}(d_i)]$-coin, using a Bernoulli factory for division to produce a $\frac{1}{(1+4b)\Pr[B_{i,j}(d_i)]}$-coin results in an exponential blow-up in computation, since the factory for the $k$-th coin would need to also simulate the factory for the $(k-1)$-st coin, and so on. Instead, we approximate these probabilities, sequentially, using multiple experiments and bounding the error using Chernoff bounds.

In order to decide whether to select some element $e_{i,j} \in S_\ell$, we repeatedly simulate our algorithm until element $e_{i,j}$, for $T = \frac{1}{2\epsilon^2} \log \frac{2nm}{\delta}$ repetitions, where the choice of running the "light scheme" and $d_i$ are fixed. In this simulation, the coins needed to make decisions until element $e_{i,j}$ are replaced with estimated coins (described shortly). Let $X_t$ be the indicator random variable for the event that $B_{i,j}(d_i)$ occurred at simulation $t \in [T]$. We select element $e_{i,j}$ (when it is active) with probability $\frac{1}{(1+4b)\left(\frac{1}{T}\sum_{t\in[T]} X_t + \epsilon\right)}$. Standard Chernoff–Hoeffding bounds [Hoe94] imply that

$$\Pr\left[\left|\frac{1}{T}\sum_{t\in[T]} X_t - \Pr[B_{i,j}(d_i)]\right| > \epsilon\right] \leq 2\exp\left(-2\epsilon^2 T\right) = 2\exp\left(-2\epsilon^2 \frac{1}{2\epsilon^2}\log\frac{2nm}{\delta}\right) \leq \frac{\delta}{nm}.$$

Assuming that $\left|\frac{1}{T}\sum_{t\in[T]} X_t - \Pr[B_{i,j}(d_i)]\right| \leq \epsilon$, $\left(\frac{1}{T}\sum_{t\in[T]} X_t + \epsilon\right) \in [\Pr[B_{i,j}(d_i)], \Pr[B_{i,j}(d_i)] + 2\epsilon]$.

When bounding $\mathbb{E}\left[W_{i,j}(d_i)\right]$ (in the proof of Theorem 3), the term $\frac{1}{(1+4b)\Pr[B_{i,j}(d_i)]} \cdot \Pr[B_{i,j}(d_i)]$ is replaced with $\frac{\Pr[B_{i,j}(d_i)]}{(1+4b)\left(\frac{1}{T}\sum_{t\in[T]}X_t+\epsilon\right)}$, which is at most $\frac{1}{1+4b}$. Thus $\mathbb{E}\left[W_{i,j}(d_i)\right] \leq \frac{2b}{1+4b}K$, which gives us $\Pr[B_{i,j}(d_i)] \geq \frac{1}{1+4b} \geq 1/5$ using the same arguments presented in the proof of Theorem 3. Thus, the probability that an active, light element $e_{i,j}$ is selected is

$$\frac{\Pr[B_{i,j}(d_i)]}{(1+4b)\left(\frac{1}{T}\sum_{t\in[T]}X_t+\epsilon\right)} \geq \frac{1}{1+4b}\left(\frac{\Pr[B_{i,j}(d_i)]}{\Pr[B_{i,j}(d_i)]+2\epsilon}\right) \geq \frac{1}{1+4b}\left(\frac{1}{1+10\epsilon}\right).$$

Using a union bound we have that

$$\Pr\left[\exists(i,j)\in[n]\times[m]: \left|\frac{1}{T}\sum_{t\in[T]}X_t - \Pr[B_{i,j}(d_i)]\right| > \epsilon\right] \leq \delta.$$

Thus, we overall have that with probability at least $1-\delta$, when we run the light scheme, each active element will be selected with probability at least $\frac{1}{1+4b}\left(\frac{1}{1+10\epsilon}\right)$. $\qquad\square$

*Proof of Proposition 2.* Notice again that, similarly to Proposition 1, the only step we cannot directly implement from the procedure outlined in the proof of Theorem 4 is the toss of a $\Pr[C_{i,j+1}(d_i)\cap B_i]$-coin. Specifically, in the proof of Theorem 4 we do not calculate the following probability exactly:

$$\Pr[C_{i,j+1}(d_i)\cap B_i] = \Pr[B_i]\Pr[C_{i,j+1}(d_i)|B_i] = \Pr[B_i] - \frac{b}{1+3b}\sum_{j'<j:j'\in S_l}x_{i,j}(d_i).$$

Our procedure will again sequentially approximate these probabilities using multiple experiments, and bounding the error using Chernoff-Hoeffding bounds.

In order to decide whether to select some element $e_{i,j}$, we repeatedly simulate our algorithm until element $e_{i,j}$, for $T = \frac{1}{2\epsilon^2}\log\frac{2nm}{\delta}$ repetitions, where the choice of running the "light scheme" and $d_i$ are fixed. Let $X_t$ be the random variable that indicates if $B_i$ occurred at simulation $t\in[T]$; from Chernoff–Hoeffding bounds [Hoe94] we have that $\Pr\left[\left|\frac{1}{T}\sum_{t\in[T]}X_t - \Pr[B_{i,j}(d_i)]\right| > \epsilon\right] \leq \frac{\delta}{nm}$. Instead of selecting element $e_{i,j}$ (when it is active) with probability $\frac{1}{(1+3b)\Pr[B_i\cap C_{i,j}(d_i)]}$, we select it with probability $\frac{1}{(1+3b)(\ell+\epsilon)}$-coin where $\ell = \frac{1}{T}\sum_{t\in[T]}X_t - \frac{b}{1+3b}\sum_{j'<j:j'\in S_l}x_{i,j}(d_i)$.

Assuming that $\left|\frac{1}{T}\sum_{t\in[T]}X_t - \Pr[B_i]\right| \leq \epsilon$ then $(\ell+\epsilon) \in [\Pr[C_{i,j+1}(d_i)\cap B_i], \Pr[C_{i,j+1}(d_i)\cap B_i]+2\epsilon]$. Thus $\frac{\Pr[C_{i,j+1}(d_i)\cap B_i]}{(1+3b)(\ell+\epsilon)} \leq \frac{1}{1+3b}$. Thus $\mathbb{E}\left[W_i\right] \leq \frac{b}{1+3b}K$ which gives us $\Pr[B_i] \geq \frac{1+b}{1+3b}$ and thus $\Pr[C_{i,j+1}(d_i)\cap B_i] \geq \frac{1}{1+3b} \geq 1/4$ using the same arguments presented in the proof of Theorem 4.

Thus,

$$\frac{\Pr[C_{i,j+1}(d_i)\cap B_i]}{(1+3b)(\ell+\epsilon)} \geq \frac{1}{1+4b}\left(\frac{\Pr[C_{i,j+1}(d_i)\cap B_i]}{\Pr[C_{i,j+1}(d_i)\cap B_i]+2\epsilon}\right) \geq \frac{1}{1+4b}\left(\frac{1}{1+8\epsilon}\right).$$

Union bounding we have that

$$\Pr\left[\exists(i,j)\in[n]\times[m]: \left|\frac{1}{T}\sum_{t\in[T]}X_t - \Pr[B_i]\right| > \epsilon\right] \leq \delta.$$

Thus with probability at least $1-\delta$ when we run the light scheme, each active element will be selected with probability at least $\frac{1}{1+4b}\left(\frac{1}{1+8\epsilon}\right)$. $\qquad\square$

# F   Extensions to procurement auctions

In this section, we show how to extend our framework for the case of procurement auctions. We only show how our framework works for sequential procurement auctions, where we construct an

auction using an OCRS (and interim allocations/payments); our results can be extended to give non-sequential auctions using a CRS, similarly to our results in Section 3.

Budget feasible procurement auctions were introduced by the seminal work of [Sin10]. Following this, there has been a line of work studying deterministic and randomized budget feasible mechanisms that obtain approximately optimal welfare, where a major focus has been on single dimensional and prior-free settings [CGL11, GJLZ20, AGN18, KS22]. We start by defining the procurement problem we study.

**Procurement Preliminaries**   There is a single buyer and a set of $n$ sellers. Each seller has a total of $m$ services they can provide. The buyer has a publicly known value $v_{i,j}$ for getting the $j$-th service that seller $i$ offers. An (integral) allocation $x \in \{0,1\}^{nm}$ indicates which services the buyer received: $x_{i,j} \in \{0,1\}$ is the indicator for whether the buyer received the $j$-th service of seller $i$. The buyer's value for an allocation $x$ is $\sum_{i \in [n], j \in [m]} x_{i,j} v_{i,j}$. The buyer will pay the sellers for their services; the buyer's objective is to maximize her value without paying more than a (publicly known) budget $B$. Each seller $i \in [n]$ has some cost for providing service $j \in [m]$ depicted as $c_{i,j}$. We will assume that $c_i$ is drawn from a known distribution $\mathcal{C}_i$; we allow for correlation between the cost for different services for a fixed seller, but require independence between sellers.

A procurement auction elicits reported costs $(c_1, \ldots, c_n)$, and determines which services are procured from which seller, as well as the payments to the sellers. Our goal is to design BIC-IR, budget-feasible procurement auctions that maximize the buyer's expected value. The definition of BIC-IR, approximate optimality, sequentiality, and interim allocations/payments are similar to the corresponding definitions from Section 2.

A procurement auction elicits reported costs $(c_1, \ldots, c_n)$, and determines which services are procured from which seller, as well as the payments to the sellers. A seller's objective is to maximize her expected utility, which is the total payment to her, minus the total cost she has to pay. A procurement auction is *Bayesian Incentive Compatible (BIC)* if every seller $i \in [n]$ maximizes her expected utility by reporting her true costs $c_i$, assuming other sellers do so as well, where this expectation is over the randomness of other sellers' valuations, as well as the randomness of the mechanism. A mechanism is *Bayesian Individually Rational* (BIR) if every seller $i \in [n]$ has non-negative expected utility when reporting her true cost (assuming other sellers do so as well). The (expected) value of a BIC procurement auction is the expected value the buyer makes when sellers draw their costs from $\mathcal{C}$ (and report their true costs to the auction). We say that a procurement auction is BIC-IR if it is both BIC and BIR. A procurement auction is *sequential* if it sequentially approaches each seller $i$, elicits a report, determines payments to seller $i$, and which services to procure from $i$, before proceeding to the next bidder. The *optimal procurement auction* for a given distribution $\mathcal{C}$, maximizes expected value over all BIC-IR procurement auctions. A procurement auction guarantees an $\alpha \geq 1$ approximation to the optimal value if its expected value is at least the expected value of the optimal procurement auction times $\frac{1}{\alpha}$.

The *interim allocation* of a procurement auction $\mathcal{M}$, $\pi^{\mathcal{M}}$, indicates, for each seller $i$ and service $j$ the probability $\pi_{i,j}^{\mathcal{M}}(r_i)$ that seller $i$ receives service $j$ when she reports cost $r_i$ (over the randomness in $\mathcal{M}$ and the randomness in other sellers' reported costs $c_{-i}$, drawn from $\mathcal{C}_{-i}$). The *interim payment* of the buyer to seller $i$, $q_i^{\mathcal{M}}(r_i)$, is the expected payment she gets when she reports cost $r_i$ (again, over the randomness in $\mathcal{M}$ and the randomness in other sellers' reported costs). The expected utility of seller $i$ with cost $c_i$ when reporting $r_i$ to a procurement auction $\mathcal{M}$, is $-\sum_{j \in [m]} c_{i,j} \pi_{i,j}^{\mathcal{M}}(r_i) + q_i^{\mathcal{M}}(r_i)$. An interim allocation rule $\pi$ is feasible in expectation if (i) $\forall i \in [n], c_i \in supp(\mathcal{C}_i), \pi_i(c_i) \in [0,1]$, and (ii) $\forall i \in [n], j \in [m], \sum_{c_i \in supp(\mathcal{C}_i)} \Pr[c_i] \cdot \pi_{i,j}(c_i) \leq 1$.

For ease of exposition, we will assume that there are no constraints on the services we can acquire, other than the buyer's budget constraint. If additional constraints exist, our framework can be extended using the ideas analyzed in the previous sections.

## F.1   Procurement Framework

Our procurement framework uses OCRSs for Stochastic Knapsack. A $c$-selectable OCRS for Stochastic Knapsack $\mu_K$ is parameterized by a knapsack size $K$ and distributions from which elements' weight are drawn. The OCRS is given, in an online manner, elements and their weight (which is drawn from the aforementioned distributions), one at a time, and it needs to decide,

immediately and irrevocably, whether to include an element in the knapsack, in a way that every element is selected with (ex-ante) probability at least $c$; see Appendix F.3 for more details.

The input to our framework is (i) a feasible in expectation, BIC-IR interim allocation, payment rule pair $(\pi, q)$ that is an $\alpha \geq 1$ approximation, (ii) a $c$-selectable OCRS for Stochastic Knapsack, and (iii) a parameter $\epsilon \geq 0$. Our framework, Algorithm 4, outputs a BIC-IR procurement auction that is a $\alpha/(c - \epsilon)$ approximately optimal.

Algorithm 4 works as follows. We approach each seller $i$ sequentially. Seller $i$ reports $r_i^* \in supp(\mathcal{C}_i)$, and we query the OCRS on input $q_i(r_i^*)$. If the OCRS selects an element with weight $q_i(r_i^*)$, we pay seller $i$ an amount equal to $q_i(r_i^*)$, with a certain probability (this step ensures that the expected payment to seller $i$ is exactly $(c - \epsilon)q_i(r_i^*)$). Finally, for each service $j \in [m]$, we receive the service from seller $i$ with probability $(c - \epsilon)\pi_{i,j}(r_i^*)$.

---

**ALGORITHM 4:** Our sequential procurement auction when given an OCRS

**Input:** an interim allocation, payment rule pair $(\pi, q)$, a $c$-selectable OCRS $\mu_B$ for Stochastic Knapsack, a parameter $\epsilon \geq 0$.

---

**for** *each seller $i \in [n]$* **do**
    Seller $i$ reports $r_i^* \in supp(\mathcal{C}_i)$.
    $Z_i \leftarrow \mu_B(q_i(r_i^*))$.
    **if** $Z_i = 1$ **then**
        Pay seller $i$, $q_i(r_i^*)$ with probability $(c - \epsilon)/p_i^*(r_i^*)$, where $p_i^*(r_i^*)$ be the probability that the OCRS selects an element with weight $q_i(r_i^*)$.
    **end**
    **for** *each service $j \in [m]$* **do**
        Receive service $j$ from seller $i$ with probability $(c - \epsilon)\pi_{i,j}(r_i^*)$
    **end**
**end**

---

**Theorem 5.** *Given (i) feasible in expectation, BIC-IR interim allocation and payment rules $(\pi, q)$ that are an $\alpha \geq 1$ approximation, (ii) a $c$-selectable OCRS for Stochastic Knapsack, and (iii) a parameter $\epsilon \geq 0$, Algorithm 4 gives a BIC-IR sequential procurement auction that is $\alpha/(c - \epsilon)$-approximately optimal. If we have query access to the OCRS, our mechanism can be implemented using a $O(poly(\sum_i |supp(\mathcal{C}_i)|, m, \frac{1}{\epsilon}))$ number of queries in expectation.*

*Proof of Theorem 5.* First, we argue that Algorithm 4 is budget feasible with probability 1. By definition, and assuming truthful reports, the interim payment $q$ defines a feasible distribution of "weights" for each seller. The OCRS always selects a set of elements whose weight is at most the knapsack size (in our case $B$), and our total payments are at most the total weight that the OCRS packs in the knapsack; therefore, our total payments are at most $B$.

Second, we argue that Algorithm 4 is BIC-IR. From the perspective of seller $i$, a report $r_i \in \mathcal{V}_i$ costs $\frac{c - \epsilon}{p_i^*(r_i)} p_i^*(r_i) q_i(r_i) = (c - \epsilon)q_i(r_i)$. The expected cost of services is $\sum_{j \in [m]} c_{i,j}(c - \epsilon)\pi_{i,j}(r_i)$. Therefore, her expected utility is $(c - \epsilon)\left(q_i(r_i) - \sum_{j \in [m]} c_{i,j}\pi_{i,j}(r_i)\right)$; since $(\pi, q)$ is BIC, so is Algorithm 4. Near-identical arguments imply the BIR guarantee.

The expected value of the buyer is $\sum_{i \in [n]} \sum_{c_i \in \mathcal{C}_i} \Pr[c_i] \sum_{j \in [m]} v_{i,j} (c - \epsilon)\pi_{i,j}(c_i)$, which is a $\alpha/(c - \epsilon)$ approximation, since $(\pi, q)$ is an $\alpha$ approximation.

If we are given only black-box access to an OCRS for Stochastic Knapsack, it is not immediately straightforward how to flip a coin with probability $(c - \epsilon)/p_i^*(r_i^*)$ (efficiently or otherwise), as needed in Algorithm 4. Using a Bernoulli factory for division (such as the result of [Mor21] discussed in Section 2), we can implement this step with $O(\frac{1}{\epsilon})$ calls in expectation; we discuss efficient implementation considerations in Appendix F.2 $\qquad \square$

[JMZ22] give a $\frac{1}{3+e^{-2}}$-selectable OCRS for Stochastic Knapsack. Combining with Theorem 5 we readily get the following application.

**Application 4** (Theorem 5 and [JMZ22]). *Consider the problem of purchasing $m$ services from $n$ strategic sellers, subject to a budget constraint. There exists a sequential, budget-feasible, BIC-IR*

*procurement auction that guarantees a $3 + e^{-2}$ ($\approx 3.13$) approximation to the expected value of the optimal BIC-IR auction.*

In Appendix F.3 we give a new OCRS for Stochastic Knapsack that is $\max\{\frac{1-k^*}{2-k^*}, \frac{1}{6}\}$-selectable, where $k^* = \frac{1}{K} \max_{i \in [n], k_i \in supp(\mathcal{K}_i)} k_i$, and $\mathcal{K}_i$ is the distribution of weights for the $i$-th element. Our OCRS outperforms the OCRS of [JMZ22] when $k^*$ is small (specifically, $k^* \le 1/3$). Furthermore, our OCRS induces a greedy and monotone OCRS for the non-stochastic knapsack problem, which is not true for the OCRS of [JMZ22]. Note that, our OCRS also implies that better approximation guarantees are possible for sequential procurement auctions if the payment to a seller is never more than a third of the total budget.

## F.2 Implementation Considerations

Here, we highlight some implementation details. First, we give a simple LP that computes optimal ($\alpha = 1$), feasible in expectation, BIC-IR interim allocation and payment rule ($\pi, q$). Second, we flesh out implementation details regarding flipping a $(c - \epsilon)/p_i^*(c_i)$-coin, when given only black-box access to an OCRS.

**Finding feasible in expectation, optimal interim rules**   Consider the following linear program, (F.2), which computes an interim relaxation of the revenue optimal BIC-IR mechanism.

$$\text{maximize} \quad \sum_{i \in [n]} \sum_{c_i \in \mathcal{C}_i} \Pr[c_i] \sum_{j \in [m]} \pi_{i,j}(c_i) v_{i,j}$$

$$\text{s.t.} \quad q_i(c_i) - \sum_{j \in [m]} c_{i,j} \pi_{i,j}(c_i) \ge q_i(c_i') - \sum_{j \in [m]} c_{i,j} \pi_{i,j}(c_i') \quad \forall i \in [n], c_i, c_i' \in supp(\mathcal{C}_i)$$

$$q_i(c_i) - \sum_{j \in [m]} c_{i,j} \pi_{i,j}(c_i) \ge 0 \qquad \forall i \in [n], c_i \in supp(\mathcal{C}_i) \tag{LP2}$$

$$\sum_{i \in [n]} \sum_{c_i \in \mathcal{C}_i} \Pr[c_i] q_i(c_i) \le B$$

$$q_i(c_i) \qquad\qquad\qquad \le B \qquad \forall i \in [n], c_i \in supp(\mathcal{C}_i)$$

This LP has $O(n \sum_{i \in [n]} |supp(\mathcal{C}_i)|)$ variables, and $O(n \sum_{i \in [n]} |supp(\mathcal{C}_i)|^2)$ constraints, and is therefore efficiently computable by standard LP solvers.

**Flipping a coin.**   We again use a Bernoulli factory for division to produce a $(c - \epsilon)/p_i^*(c_i)$-coin. $c - \epsilon$ is known. And, identically to our approach in Section 3.1, we can flip a $p_i^*(c_i)$-coin by simulating the entire procedure, conditioning on $c_i$ being the report of seller $i$.

## F.3 A new OCRS for Stochastic Knapsack

In the Stochastic Knapsack problem, there is a ground set of elements $N = \{e_i\}_{i \in [n]}$ and a knapsack size $K$. Each element arrives sequentially and reveals a random weight $k_i \in [0, K]$ drawn from a known prior distribution $\mathcal{K}_i$ (where a draw of $k_i = 0$ is analogous to element $e_i$ being inactive/not arriving). The input distribution satisfies $\sum_{i \in [n]} \sum_{k_i \in supp(\mathcal{K}_i)} \Pr[\mathcal{K}_i = k_i] \cdot k_i \le K$. Once an element arrives and reveals its weight we need to immediately and irrevocably decide whether this element is included in the knapsack. A $c$-selectable OCRS for this problem is a procedure that selects elements (online), such that the knapsack constraint is never violated (i.e., $\sum_{i \in [n]} k_i \le K$ in all outcomes), and every element is selected with probability at least $c$.

**Theorem 6.** *There exists a $\max\{\frac{1-k^*}{2-k^*}, \frac{1}{6}\}$-selectable OCRS for Stochastic Knapsack, where $k^* = \frac{1}{K} \max_{i \in [n], k_i \in supp(\mathcal{K}_i)} k_i$.*

*Proof of Theorem 6.* We present two OCRSs: a $\gamma$-selectable OCRS, where $\gamma = \frac{1-k^*}{2-k^*}$, and a $\frac{1}{6}$-selectable OCRS. Our overall OCRS computes $k^* = \frac{1}{K} \max_{i \in [n], k_i \in supp(\mathcal{K}_i)} k_i$. If $\gamma \ge \frac{1}{6}$, it executes the following $\gamma$-selectable OCRS; otherwise, it executes a $\frac{1}{6}$-selectable OCRS.

First, we give the $\gamma$-selectable OCRS. Initialize $I = \emptyset$, and let $C_i(k_i)$ be the event that $\sum_{i' \in I} k_{i'} \leq K - k_i$ when element $e_i$ arrives, and given that its weight is $k_i$. When element $e_i \in N$ with weight $k_i$ arrives, if $\sum_{i' \in I} k_{i'} \leq K - k_i$, we include $e_i$ in $I$ with probability $\frac{\gamma}{\Pr[C_i(k_i)]}$, where $\gamma = \frac{1-k^*}{2-k^*}$. The probability with which an element $e_i$ is selected is then:

$$\sum_{k_i \in supp(\mathcal{K}_i)} \Pr[\mathcal{K}_i = k_i] \Pr[C_i(k_i)] \frac{\gamma}{\Pr[C_i(k_i)]} = \gamma \sum_{k_i \in supp(\mathcal{K}_i)} \Pr[\mathcal{K}_i = k_i] = \gamma.$$

It remains to prove that $\frac{\gamma}{\Pr[C_i(k_i)]}$ is a valid coin (i.e., $\Pr[C_i(k_i)] \geq \gamma$). Let $W_i$ be the random variable that represents the total weight of elements in $I$ (i.e. $\sum_{i' \in I} k_{i'}$) when element $i$ arrives.

$$\mathbb{E}[W_i] = \sum_{i' < i} \sum_{k_{i'} \in supp(\mathcal{K}_{i'})} \Pr[C_{i'}(k_i)] \frac{\gamma}{\Pr[C_{i'}(k_i)]} \Pr[\mathcal{K}_{i'} = k_{i'}] k_{i'} \leq \gamma K.$$

Therefore, we have

$$\begin{aligned}
\Pr[C_i(k_i)] &= \Pr[E_i \leq K - k_i] \\
&= 1 - \Pr[W_i > K - k_i] \\
&\geq 1 - \frac{\gamma K}{K - k_i} \qquad &\text{(Markov's Inequality)} \\
&\geq 1 - \frac{\gamma}{1 - k^*} \\
&= \gamma.
\end{aligned}$$

This concludes the proof for the $\gamma$-selectable OCRS.

Next, we give a $\frac{1}{6}$-selectable OCRS. With probability $1/2$ we run a "heavy scheme," that only considers elements $e_i$ such that $k_i > \frac{K}{2}$; otherwise, we run a "light scheme," that only considers elements $e_i$ such that $k_i \leq \frac{K}{2}$.

Suppose we run the heavy scheme. Initialize $I = \emptyset$, and let $A_i$ be the event that $I = \emptyset$ when element $e_i$ arrives. For each element $e_i$ such that $k_i > \frac{K}{2}$, if $I = \emptyset$, we select $e_i$ with probability $\frac{1}{3\Pr[A_i]}$. Assuming that $\frac{1}{3\Pr[A_i]}$ is a valid coin (i.e., $\Pr[A_i] \geq 1/3$), the probability with which each element is selected, given that it is heavy and that we run the heavy scheme, is $\Pr[A_i] \frac{1}{3\Pr[A_i]} = 1/3$. To prove that $\frac{1}{3\Pr[A_i]}$ is a valid coin we have:

$$\begin{aligned}
\Pr[A_{i+1}] &= \Pr[A_i] \left( 1 - \frac{1}{3\Pr[A_i]} \sum_{k_i > K/2} \Pr[\mathcal{K}_i = k_i] \right) \\
&= \Pr[A_i] - \frac{1}{3} \sum_{k_i > K/2} \Pr[\mathcal{K}_i = k_i] \\
&= 1 - \frac{1}{3} \sum_{i' \leq i} \sum_{k_i > K/2} \Pr[\mathcal{K}_i = k_i] \\
&\geq 1 - 2/3 = 1/3.
\end{aligned}$$

Now, suppose we run the light scheme. Notice that in this regime, where we ignore elements whose weight is larger than $K/2$, the previous $\gamma$-selectable OCRS is $1/3$-selectable (since $k^* = \frac{1}{K} \max_{i \in [n], k_i \in supp(\mathcal{K}_i)} k_i$). Since each scheme (heavy and light) is chosen with probability $1/2$, this OCRS is $1/6$-selectable.

This concludes the proof for the $\frac{1}{6}$-selectable OCRS. $\qquad \square$

**Proposition 3.** *We can implement a $\left( c \left( \frac{1-\delta}{1+2\,\epsilon/c} \right) \right)$-selectable OCRS for the Stochastic Knapsack setting in time $poly(1/\epsilon^2, \log(1/\delta), n)$, where $c = \max\{ \frac{1-k^*}{2-k^*}, 1/6 \}$, and $k^* = \frac{1}{K} \max_{i \in [n], k_i \in supp(\mathcal{K}_i)} k_i$.*

*Proof of Proposition 3.* The only step we cannot directly implement from the procedure outlined in the proof of Theorem 6 is the toss of the $\frac{\gamma}{\Pr[C_i(k_i)]}$ coin. The use of a Bernoulli factory would exponentially blow up the complexity of the procedure. Instead, we approximate these probabilities, sequentially, using multiple experiments and bounding the error using Chernoff bounds.

In order to decide whether to select some element $e_i \in N$, we repeatedly simulate our algorithm until element $e_i$, for $T = \frac{1}{2\epsilon^2} \log \frac{2|N|}{\delta}$ repetitions. In this simulation, the coins needed to make decisions until element $e_i$ are replaced with estimated coins (described shortly). Let $X_t$ be the indicator random variable for the event that $C_i(k_i)$ occurred at simulation $t \in [T]$. Instead of selecting element $e_i$ (when it is active) with probability $\frac{\gamma}{\Pr[C_i(k_i)]}$, we select it with probability $\frac{\gamma}{\left(\frac{1}{T}\sum_{t\in[T]} X_t + \epsilon\right)}$. Standard Chernoff–Hoeffding bounds [Hoe94] imply that

$$\Pr\left[\left|\frac{1}{T}\sum_{t\in[T]} X_t - \Pr[C_i(k_i)]\right| > \epsilon\right] \leq 2\exp\left(-2\epsilon^2 T\right) = 2\exp\left(-2\epsilon^2 \frac{1}{2\epsilon^2}\log\frac{2|N|}{\delta}\right) \leq \frac{\delta}{|N|}.$$

Assuming that $\left|\frac{1}{T}\sum_{t\in[T]} X_t - \Pr[C_i(k_i)]\right| \leq \epsilon$, $\left(\frac{1}{T}\sum_{t\in[T]} X_t + \epsilon\right) \in [\Pr[C_i(k_i)], \Pr[C_i(k_i)] + 2\epsilon]$.

When bounding $\mathbb{E}[W_i]$ (in the proof of Proposition 3), the term $\frac{\gamma}{\Pr[C_i(k_i)]} \cdot \Pr[C_i(k_i)]$ is replaced with $\frac{\gamma \Pr[C_i(k_i)]}{\left(\frac{1}{T}\sum_{t\in[T]} X_t + \epsilon\right)}$, which is at most $\gamma$. Thus $\mathbb{E}[W_i] \leq \gamma K$, which gives us $\Pr[C_i(k_i)] \geq \gamma$ using the same arguments presented in the proof of Proposition 3. Thus, the probability that an active, element $e_i$ is selected is

$$\frac{\Pr[\gamma C_i(k_i)]}{\left(\frac{1}{T}\sum_{t\in[T]} X_t + \epsilon\right)} \geq \gamma\left(\frac{\Pr[C_i(k_i)]}{\Pr[C_i(k_i)] + 2\epsilon}\right) \geq \gamma\left(\frac{1}{1 + 2\epsilon/\gamma}\right).$$

Using a union bound we have that

$$\Pr\left[\exists i \in N : \left|\frac{1}{T}\sum_{t\in[T]} X_t - \Pr[C_i(k_i)]\right| > \epsilon\right] \leq \delta.$$

Thus, we overall have that with probability at least $1 - \delta$, when we run the light scheme, each active element will be selected with probability at least $\gamma\left(\frac{1}{1+2\epsilon/\gamma}\right)$. $\qquad\square$

