# OpenReview forum: "Mechanism Design via the Interim Relaxation"
_NeurIPS.cc/2025/Conference — NeurIPS 2025 poster_

### Official Review · Reviewer_hY62 · 2025-06-09

**Clarity:** 3
**Significance:** 3
**Originality:** 4
**Rating:** 5
**Confidence:** 3

**Summary:**

The paper suggests a new method, called two-level (O)CRS, that allows for "Vertical-horizontal" constraints (i.e., matroid constraints over both which items each agent gets, and which agents an item is allocated to). The paper shows how existing (O)CRSs can be combined into a t(O)CRS, and apply it to some settings, most notably improving [Cai & Zhao '17]'s approximation of 70 for the optimal revenue of an auction for constrained additive bidders, to around 3.16.

**Questions:**

Q1: I didn’t manage to understand the motivation for the vertical-horizontal constraints. I understand why the items allocated to an agent will have matroid constraints. But what does it mean that the agents that get item j satisfy matroid constraints? Isn’t it always Ex-post allocated to a single agent? Please explain the settings, and also references to works that previously discussed it before could be helpful.
I see later that you mention [14] that restrict each item to be allocated to at most one agent: So maybe another way to ask my question is, so isn’t this guarantee already built into the CRS framework as the feasibility constraint (I guess not)?

Q2: What is the argument for the last inequality in 1112? Why is the sum of probabilities for these elements to be active less than 4b (I can understand where the b is coming from, but why 4)? Same question for the last inequality of 1123, where it seems you use the fact that the sum of all preceding k_{i,j} are less than K/2, and again show a bound of 4 for the same sum of probabilities (but somehow here b is already outside the parantheses? so shouldn’t it be 2b^2/(1+4b)?)

**Ethical Concerns:**

["NO or VERY MINOR ethics concerns only"]

**Final Justification:**

I was in support of accepting the paper and remain so. I had two minor questions and the rebuttal addressed them promptly. The other reviewers seem to lean towards acceptance as well, though they raise an issue of whether the algorithms are complicated for practical application.

**Limitations:**

Yes

**Quality:**

4

**Strengths And Weaknesses:**

Strengths:

- The technical contribution is significant, and in some cases the results improve over the state of the art.

- The paper is well-written

- Personally I found Section B very helpful, as it provided a good introduction to the topic of Bernoulli factories.

Weaknesses:

- Since the topic is quite heavy and I'm not familiar with it, even though the paper is clearly written, there were still some things I couldn't figure out, which I detail here and in the questions:

Corollary 1: It’s not clear from the statement of Theorem 2 that if you combine a CRS over items constraints with an OCRS over agent constraints you get a tOCRS (rather than tCRS). By the statement it seems like if you combine two CRSs, you get a tCRS, and if you combine two OCRSs, you get a tOCRS.

Notice that in the proof of Theorem 1 you didn’t explicitly discuss the revenue approximation guarantee, but it’s pretty clear given the proof.

---

> ### Author Rebuttal · Authors · 2025-07-30
>
> Thank you for your review.
>
> >Corollary 1: It’s not clear from the statement of Theorem 2 that if you combine a CRS over items constraints with an OCRS over agent constraints you get a tOCRS (rather than tCRS). By the statement it seems like if you combine two CRSs, you get a tCRS, and if you combine two OCRSs, you get a tOCRS.
>
> This is a fair question. Indeed, combining a CRS over item constraints with an OCRS over agent constraints *does* yield a valid tOCRS. We agree that this may not be completely transparent from the high-level statement, and we’ll clarify this.
>
> To explain, it’s simple to present the argument from a mechanism design perspective. Each agent arrives online and declares their $d_i$ ($d_i$ from Definition 4). The mechanism needs to decide which items to give them so that the final allocation is *always* feasible. We can run a CRS on the ``agent constraint,’’ since all the elements for that constraint correspond to items that will be allocated to that agent, and are all items already present. At the same time, we can independently run an OCRS per item with the agent constraints. We allocate the item to this agent only if **both** the OCRS and CRS agree to this allocation. Since for a particular element/item $e_{i,j}$ the $i$-th CRS and the $j$-th OCRS are oblivious of each other, with separate randomness, the probability that both are active is simply the product of the probabilities of being active in each process. The resulting procedure is a valid tOCRS by definition.
>
> We’ll add a sentence or two to Theorem 2 to make this clearer in the final version.
>
> >I didn’t manage to understand the motivation for the vertical-horizontal constraints. I understand why the items allocated to an agent will have matroid constraints. But what does it mean that the agents that get item j satisfy matroid constraints? Isn’t it always Ex-post allocated to a single agent? Please explain the settings, and also references to works that previously discussed it before could be helpful. I see later that you mention [14] that restrict each item to be allocated to at most one agent: So maybe another way to ask my question is, so isn’t this guarantee already built into the CRS framework as the feasibility constraint (I guess not)?
>
>
> VH constraints are feasibility constraints for each item (across agents) and **separately** for each agent (across items). So, we have constrains on which items can go to a single agent $i$ --- e.g., “agent $i$ can get at most $k$ items” --- and constraints on how a single item $j$ can be allocated across agents --- e.g., “there are $k$ copies of item $j$, so item $j$ can be allocated to at most $k$ agents” (so, indeed, we allow for an item to be allocated to multiple agents). Perhaps the phrasing of our setup (line 150 “...$m$ indivisible, heterogeneous items…”) is what caused the confusing regarding the second point; we can rephrase.
>
>
> >What is the argument for the last inequality in 1112? Why is the sum of probabilities for these elements to be active less than 4b (I can understand where the b is coming from, but why 4)?
>
> Thank you for catching this! In the second to last equality there is a $b$ missing outside the parenthesis, which we will update. Intuitively, from the perspective of any given agent $i$, an element $e_{i’, j’}$ is active with probability $b w_{i’,j’}$ (they do not know $d_{i’}$), and an element $e_{i,j’}$ is active with probability $b x_{i,j’}(d_i)$.
>
> Regarding the “4” factor: from the first condition in Definition 4, and since heavy items cannot weigh less than $K/2$, we have that the sum of $x_{i,j}(d_i)$ cannot be more than $2$. Similarly, from the second condition of Definition 4, and since the heavy items cannot weigh less than $K/2$, we have that the sum of $w_{i,j}$s cannot be more than $2$ as well. Combining we get the $4$. This is tight: there might have two items that weigh $K/2 + \epsilon$ each and the agent almost always receives both.
>
> >Same question for the last inequality of 1123, where it seems you use the fact that the sum of all preceding k_{i,j} are less than K/2, and again show a bound of 4 for the same sum of probabilities (but somehow here b is already outside the parantheses? so shouldn’t it be 2b^2/(1+4b)?)
>
> There is no extra $b$ term (the omission was the mistake earlier). As for the $4$ bound, the argument is similar: from Definition 4 we have that the expected weight of elements that are active before agent $i$’s turn cannot be more than $K$, and the expected weight of elements that are active in $i$’s round cannot be more than $K$ as well. Combining the two inequalities we get the desired 2K bound.

---

> > ### Comment · Reviewer_hY62 · 2025-08-07
> > **Ack**
> >
> > I thank the authors for their rebuttal and remain in support of accepting the paper.

---

### Official Review · Reviewer_9suo · 2025-06-30

**Clarity:** 2
**Significance:** 3
**Originality:** 4
**Rating:** 4
**Confidence:** 3

**Summary:**

This paper studies the problem of designing revenue-maximizing mechanisms for selling m heterogeneous items to
n additive agents, subject to downward-closed feasibility constraints. The authors propose a framework based on the following three steps:

- Relaxing the feasibility constraints to allow interim allocation/payment rules that only satisfy the constraints in expectation.

- Solving a linear program to compute an optimal Bayesian incentive-compatible (BIC) and individually rational (IR) interim solution.

- Rounding the fractional solution using an extension of contention resolution scheme (OCRS) techniques.

The main contribution is the introduction of a new variant of OCRS that allows dependencies between the activations of elements, broadening the applicability of the framework. They also apply their framework to obtain approximation guarantees, including:

- A 3.16-approximation sequential mechanism under “matroid Vertical-Horizontal” constraints (although the exact setting is not clearly defined—see comments).

- A sequential, efficiently computable 10-approximation mechanism for additive agents under knapsack constraints.

- A 9-approximation mechanism for agents with "arbitrary" valuations under knapsack constraints (this needs clarification—see comments).

**Questions:**

In the Applications section, when you say agents have arbitrary valuation functions, does that mean monotonicity, subadditivity, or other standard assumptions can be violated? Please clarify the assumptions.

In the first application (matroid VH constraints), can an item be allocated to multiple agents? Also, what does it mean for agents to "form a matroid"?

**Ethical Concerns:**

["NO or VERY MINOR ethics concerns only"]

**Final Justification:**

The authors have answered my questions. I think my concerns about presentation can be resolved if the paper is accepted. The paper has merits, and I am positive about its acceptance. However, I am not fully convinced that the novelty of the results and the framework is sufficient to strongly support acceptance. Therefore, my evaluation remains weak accept.

**Limitations:**

Yes.

**Paper Formatting Concerns:**

The presentation of the paper can be significantly improved. Key definitions, intuition, and algorithmic descriptions are often compressed or interleaved, which can make the material hard to follow, especially for readers less familiar with the area. While space limitations are understandable, reorganizing the exposition and including more motivating examples or intuition would go a long way in improving accessibility.
For example:

- Clarify important concepts like downward-closed valuations, ex-ante, and ex-post feasibility early in the paper.

- Section 2 ("Preliminaries") is particularly dense. It introduces many notations in rapid succession—e.g., valuation vectors, marginals, feasibility polytopes—without sufficient intuition.

- Line 154 (and before that in the our contribution section): “noting that  $D_{i,j}$ is not necessarily independent of $D_{i,j'}$” is vague. Please explain why.

- (this is only a suggestion) It would be helpful to briefly explain why it is computationally hard to reconstruct an ex-post mechanism from interim rules—perhaps with a reference or a short example.

- Additionally, when listing results (e.g., "Theorem 3, Theorem 4, and Theorem 1"), it would be better to list them in the order in which they appear in the paper.

- The bibliography formatting is inconsistent. Some references use abbreviations, some include DOIs, and others do not. Please standardize the references using a consistent citation style.

- When referring to lines in algorithms (e.g., "Line 1 of Algorithm 1", "Line 3 of Algorithm 3"), note that the algorithms are not line-numbered. Also, in Algorithm 3, “Line 3” seems to refer to the “end” command—please double-check and correct these references.

**Quality:**

3

**Strengths And Weaknesses:**

I think the extension of the (O)CRS framework to handle dependencies among elements is technically nontrivial and represents an important contribution. The framework is flexible, applies to multiple constraint types, and in some cases improves over prior results.

The presentation of the paper can be significantly improved. Key definitions, intuition, and algorithmic descriptions are often compressed or interleaved, which can make the material hard to follow, especially for readers less familiar with the area. I understand that some of these issues are due to the lack of space, but the paper as it is now is hard to follow.

---

> ### Author Rebuttal · Authors · 2025-07-30
>
> Thank you for your review. We will incorporate all reviewer feedback to improve the presentation.
>
> > In the Applications section, when you say agents have arbitrary valuation functions, does that mean monotonicity, subadditivity, or other standard assumptions can be violated? Please clarify the assumptions.”
>
> Yes, the valuation functions in this application can be arbitrary: non-monotone, subadditive, superadditive, anything works. The only constraint is that the value is non-negative for every subset of items. We will clarify this.
>
> > In the first application (matroid VH constraints), can an item be allocated to multiple agents? Also, what does it mean for agents to "form a matroid"?
>
> Allocating an item to multiple agents is allowed, if desired. It can also be disallowed, if desired. More generally, VH constraints are feasibility constraints for each item (across agents) and **separately** for each agent (across items).  For example, a cardinality constraint that allows for at most $k$ items to be allocated to an agent (e.g., if the agent is $k$-demand) is a per-agent component of a VH constraint. Many settings also require per-item constraints. For example, in limited supply markets, an item might be assigned to at most one agent, or more generally, to at most $k$ agents when $k$ copies are available. Note also that this is an example of a matroid constraint ($k$-uniform matroids). VH constraints can express a wide variety of combinatorial structures, including matroid constraints.
>
> Matroids offer a natural abstraction for various feasibility constraints. For example, transversal matroids: consider allocating houses to $n$ agents, where each agent has a set of acceptable houses; the feasible set of agents that can be allocated forms an independent set of a transversal matroid. Another example is partition and laminar matroids: consider allocating resources fairly when agents belong to different subgroups and want to avoid overallocating to a single subgroup. Resource allocation under matroid constraints is fundamental and well-studied in the literature, e.g.,:
> - Frugality of Truthful Mechanisms. Karlin, Kempe, Tamir. Beyond VCG: FOCS 2005
> - Simple versus Optimal Mechanisms. Hartline, Roughgarden. EC 2009
> - Matroids, Secretary Problems, and Online Mechanisms. Babaioff, Immorlica, Kleinberg. JACM 2018
> - Matroid Prophet Inequalities. Kleinberg, Weinberg. STOC 2012.
> - An Ascending Vickrey Auction for Selling Bases of a Matroid. Bikhchandani, de Vries, Schummer, Vohra. OR 2011.
> - Deferred-Acceptance Clock Auctions and Radio Spectrum Reallocation. Milgrom, Segal. JPE 2017.

---

> > ### Comment · Reviewer_9suo · 2025-08-05
> >
> > Thank you for your response. My questions have been answered.

---

### Official Review · Reviewer_PWNn · 2025-07-01

**Clarity:** 3
**Significance:** 2
**Originality:** 2
**Rating:** 4
**Confidence:** 3

**Summary:**

Based on the influential work of Alaei, the authors introduce a novel, general framework for designing both non-sequential and sequential multi-agent, (approximately-optimal) revenue-maximizing mechanisms for agents with additive preferences, subject to downward-closed constraints on the set of feasible allocations. The framework relies on an interim relaxation, i.e., relaxation of the feasibility constraints, resulting in interim rules that are feasible in expectation. It also features two-level (online) contention resolution schemes (termed tOCRSs), which are variants of the classic (O)CRSs, and allow for dependencies between the activation of different elements. The framework takes as input such a BIC-IR interim allocation and a tCRS/tOCRS and outputs an approximately optimal mechanism for agents with constrained additive valuations. To find interim rules that are optimal and feasible in expectation, solving a (large) linear program is required. tCRSs and tOCRSs can be constructed in a black-box manner using existing CRSs/OCRSs for a general family of constraints (termed vertical-horizontal constraints). The authors also independently give a tOCRS for Knapsack constraints and a tOCRS for Multi-Choice Knapsack constraints.

They demonstrate how their framework can construct improved or new (sequential) mechanisms in three applications, which are natural and general enough, with one of them even allowing for arbitrary valuation functions for the agents (albeit computationally inefficient). For two of the applications, they use the novel tOCRS they construct for knapsack constraints. In the appendix, they also give an extension to procurement auctions.

**Questions:**

See questions under weaknesses and some smaller ones mentioned below.

Are these constant-factor approximations for applications 2 and 3 the first for such settings in the literature? For application 2, it is mentioned that it has been studied only for the welfare objective. Are there lower bounds for these two settings?

The result for the arbitrary valuation functions is positive (despite not being computationally efficient); what makes it go beyond additive valuations and, apart from the tOCRS for knapsack, could the framework be used as a base, with new techniques on top, to go beyond additive valuations?

I’m trying to understand what VH constraints exactly express. Is there some intuition behind the definition or is the definition mostly tailored to the specific application that the authors associate with the framework? What else do they capture in terms of known settings with combinatorial constraints?

As mentioned before, the computational efficiency also depends on whether the polytopes have efficient representations for solving the LP efficiently. From well-established mechanisms in the literature or other practical examples, when is this the case?

**Ethical Concerns:**

["NO or VERY MINOR ethics concerns only"]

**Final Justification:**

The authors have answered my questions. The paper is well-written and makes a nice theoretical contribution at the intersection of the mechanism design and OCRS literature. I have some concerns about the practical implementation and it would be good to already have some indication whether the framework can be efficiently implemented in some scenarios.

**Limitations:**

yes

**Paper Formatting Concerns:**

No concerns

**Quality:**

3

**Strengths And Weaknesses:**

Strengths:

The paper makes for a nice read; the contributions (and limitations) are clearly stated and placed in the current OCRS and mechanism design literature, and the proofs are well-written.

The framework contains several nice components (finding the interim rules, defining and constructing the tCRS/tOCRS, the Bernoulli factory for the probabilities); it’s non-trivial to handle each of them separately and then bring them together. Also, the new tOCRS for knapsack constraints, can also handle mechanisms with arbitrary valuation functions, as demonstrated in Application 3, so it seems to have some nice generalization properties.

Weaknesses:

The authors provide different results and applications, but I feel without a single point standing out. For instance, the framework is nice and presented as the main contribution of the paper, but then applied only to one of the three applications, which is an interesting enough one to improve upon the constant factor from the paper of Cai and Zhao. However, as mentioned in the submission, this was a subcase in their work, so they designed a broad enough framework that captured also this setting as a special case, and it probably not their main goal to optimize for the constant factor. Could you expand a bit on the importance of the framework and other well-known settings that it improves or provides first approximation guarantees? Or how the techniques used can provide new insights for either the construction of OCRSs or the design of (sequential) allocation mechanisms for other settings that are difficult to tackle? The other two applications are built on the specialized tOCRSs, which are a nice contribution, but maybe mainly of interest for the community working on CRS/OCRS problems for mechanism design. For these reasons, maybe the scope of the paper is a bit limited for NeurIPS and would fit better in either an EconCS or TCS conference.

Computational concerns and applicability: As the authors state in 3.1, the LP is big (already the number of constraints is polynomial in the sum of the support sizes for BIC-IR + rest of polyhedral constraints). How can you deploy such schemes in practice? When all support sizes are small and there is an efficient representation of polytopes the LP step could be fast, does that occur often in practice? Could you expand a bit on how such a framework can be deployed in practice, in which situations, and if/how it can scale on its various parameters for large instances? Also, the construction of the tOCRS depends on the existence of the relevant OCRSs, which should also be easily implementable. The coin flipping processes for the non-constructive steps in the tOCRSs for knapsack might also be computationally demanding.In general, the principle is that when you go for approximately optimal mechanisms, you trade simplicity for approximation, which am not sure is happening here. For demonstrating some of the applicability of the framework and/or the tOCRSs built from scratch, it would be interesting to have some experimental results, e.g., for application 1, for the mechanism in this paper and that of Cai and Zhao (or other heuristics), to compare approximations and computational efficiency in real-world scenarios.

---

> ### Author Rebuttal · Authors · 2025-07-30
>
> Thank you for your review.
>
> > Are these constant-factor approximations for applications 2 and 3 the first for such settings in the literature? For application 2, it is mentioned that it has been studied only for the welfare objective. Are there lower bounds for these two settings?
>
> This question is also related to the first weakness in your review (“...applied only to one of the three applications…”). First, we note that all three applications use the framework. The procurement setting is also, morally, a fourth application. Second, yes, Applications 2 and 3 give the first known *sequential* mechanisms with constant-factor approximations for such general settings. For Application 1, the result closest in generality is the 70 approximation of Cai and Zhao. To avoid overstating our contribution, we would also like to note that non-sequential optimal mechanisms can be efficiently computed using the framework of Cai et al.; however, these mechanisms are neither simple nor sequential.
>
> >Or how the techniques used can provide new insights for either the construction of OCRSs or the design of (sequential) allocation mechanisms for other settings that are difficult to tackle?
>
> We agree that new techniques for constructing OCRSs are of broad interest. However, we view this as somewhat beyond the scope of our paper. Our main contribution is not a new general technique for constructing OCRSs, but rather a framework that integrates existing OCRSs into approximately optimal mechanism design, particularly in the sequential setting. That said, our tOCRSs for Knapsack and Multi-Choice Knapsack are new, and we believe this contributes meaningfully to the broader OCRS literature.
>
> >The result for the arbitrary valuation functions is positive (despite not being computationally efficient); what makes it go beyond additive valuations and, apart from the tOCRS for knapsack, could the framework be used as a base, with new techniques on top, to go beyond additive valuations?
>
> The extension to arbitrary valuations comes from creating a meta-item for each subset of items, and adding the appropriate constraint (each agent can get at most one meta-item); our framework provides enough flexibility to give a sequential mechanism in this setting. To your broader question, our framework could plausibly serve as a base for moving beyond additive valuations more generally; the bottleneck would be getting the appropriate tCRSs and tOCRSs.
>
> >I’m trying to understand what VH constraints exactly express. Is there some intuition behind the definition or is the definition mostly tailored to the specific application that the authors associate with the framework? What else do they capture in terms of known settings with combinatorial constraints?
>
> VH constraints are feasibility constraints for each item (across agents) and **separately** for each agent (across items).  For example, a cardinality constraint that allows for at most $k$ items to be allocated to an agent (e.g., if the agent is $k$-demand) is a per-agent component of a VH constraint. Many settings also require per-item constraints. For example, in limited supply markets, an item might be assigned to at most one agent, or more generally, to at most $k$ agents when $k$ copies are available. Note also that this is an example of a matroid constraint ($k$-uniform matroids). VH constraints can express a wide variety of combinatorial structures, including matroid constraints.
>
> Matroids offer a natural abstraction for various feasibility constraints. For example, transversal matroids: consider allocating houses to $n$ agents, where each agent has a set of acceptable houses; the feasible set of agents that can be allocated forms an independent set of a transversal matroid. Another example is partition and laminar matroids: consider allocating resources fairly when agents belong to different subgroups and want to avoid overallocating to a single subgroup. Resource allocation under matroid constraints is fundamental and well-studied in the literature, e.g.,:
> - Frugality of Truthful Mechanisms. Karlin, Kempe, Tamir. Beyond VCG: FOCS 2005
> - Simple versus Optimal Mechanisms. Hartline, Roughgarden. EC 2009
> - Matroids, Secretary Problems, and Online Mechanisms. Babaioff, Immorlica, Kleinberg. JACM 2018
> - Matroid Prophet Inequalities. Kleinberg, Weinberg. STOC 2012.
> - An Ascending Vickrey Auction for Selling Bases of a Matroid. Bikhchandani, de Vries, Schummer, Vohra. OR 2011.
> - Deferred-Acceptance Clock Auctions and Radio Spectrum Reallocation. Milgrom, Segal. JPE 2017.
>
> >As mentioned before, the computational efficiency also depends on whether the polytopes have efficient representations for solving the LP efficiently. From well-established mechanisms in the literature or other practical examples, when is this the case?
>
> Our LP (LP1) is significantly simpler than the LPs in the Cai et al. framework. In particular, we only require feasibility in expectation, which avoids the more complex ex-post constraints handled in their work. Is our framework simple enough to be practical as a black-box tool in practice? That depends on the application at hand. As with much of algorithmic mechanism design (including, e.g., the framework of Cai et al.), there is a tension between theoretical generality and real-world deployability. That said, many real-world mechanisms (e.g., see the recent work on autobidding) are governed less by formal polytime guarantees and more by having “clean knobs” to tune or reason about. Our framework separates the interim form of the mechanism (solution of the LP) from its ex-post implementation (tOCRSs), so our work does contribute to this agenda.

---

> > ### Comment · Reviewer_PWNn · 2025-08-05
> >
> > Thank you for your replies to my questions. Most of them are now resolved, but I would still appreciate if you could provide further clarifications on the two following points:
> >
> > - Could you expand a bit more on this sentence of your rebuttal? "That said, many real-world mechanisms (e.g., see the recent work on autobidding) are governed less by formal polytime guarantees and more by having “clean knobs” to tune or reason about." I still feel that a mechanism that requires a number of queries that is polynomial in the supports of the distributions (and other parameters), as stated in Theorem 1, can often become prohibitive to deploy in practice.
> > - I understand that all three applications use the framework, I apologize that I didn't phrase it correctly in the review. What I wanted to ask is if you have in mind another well-known/important sequential or non-sequential mechanism design setting, where if we were to find a new/improved CRS/OCRS (not sure about the current most promiment open questions in this field), it would automatically imply, by applying your framework, a new/improved mechanism (similar to what happens for application 1).

---

> > > ### Author Response · Authors · 2025-08-05
> > >
> > > Thank you for your continued engagement.
> > >
> > > > Could you expand a bit more on this sentence of your rebuttal? "That said, many real-world mechanisms (e.g., see the recent work on autobidding) are governed less by formal polytime guarantees and more by having “clean knobs” to tune or reason about." I still feel that a mechanism that requires a number of queries that is polynomial in the supports of the distributions (and other parameters), as stated in Theorem 1, can often become prohibitive to deploy in practice.
> > >
> > > Our point in the rebuttal was not to claim that runtime polynomial in the support is immediately deployable for all practical applications, but to highlight that, in practice, comparisons in terms of worst-case guarantees are not always the deciding factor in deployability. What matters just as much (if not more) in many deployed systems is the simplicity and transparency of the format of the mechanism. For instance, widely used formats like generalized second price (GSP) or uniform bidding in autobidders (see "Auto-bidding and Auctions in Online Advertising: A Survey" by Aggarwal et al. for a recent survey) are valued not for computational optimality, but because they are interpretable and compatible with existing infrastructure. Our work contributes to this agenda by demonstrating that the interim form of a mechanism suffices to derive an online mechanism in very general settings.
> > >
> > > > I understand that all three applications use the framework, I apologize that I didn't phrase it correctly in the review. What I wanted to ask is if you have in mind another well-known/important sequential or non-sequential mechanism design setting, where if we were to find a new/improved CRS/OCRS (not sure about the current most promiment open questions in this field), it would automatically imply, by applying your framework, a new/improved mechanism (similar to what happens for application 1).
> > >
> > > Thank you for the clarification. For additive agents, the mechanisms we provide are very general already, so we are not sure if new OCRSs are needed. Of course, improved OCRSs for any matroid constraint or knapsack constraint can be used in a black-box way to get new and improved mechanisms. Beyond additive agents (which is an important mechanism design problem), the problem becomes more interesting. Our intuition is that a slight change in our definitions for tOCRSs/tCRSs could extend the framework to richer valuation classes. However, getting these tOCRSs/tCRSs may not always be possible to do in a clean, black-box way, since more complicated correlations might arise (at least that's our intuition). Nevertheless, we see this as a compelling open direction.

---

> > > > ### Comment · Reviewer_PWNn · 2025-08-08
> > > >
> > > > Thank you for the additional clarifications. After the rebuttal, I appreciate more the theoretical contribution of the paper. At the same time, I still have my reservations about the practicality of the framework, since I'm not sure how some of the obstacles in the implementation could be overcome. Overall, I remain borderline, but I am happy to weakly lean towards acceptance; I will update my score accordingly.

---

### Official Review · Reviewer_y7Bn · 2025-07-02

**Clarity:** 4
**Significance:** 3
**Originality:** 3
**Rating:** 4
**Confidence:** 3

**Summary:**

The paper studies the online allocation problem, where agents report valuations for multiple heterogeneous products and make payments if a product is successfully allocated to them. The objective is to design a mechanism that guarantees Bayesian Incentive Compatibility and Individual Rationality (BIC-IR), while achieving approximate optimality up to a constant ratio. This extends existing work that simplifies the problem using single-agent reduction.

The key idea is to introduce a two-level interim relaxation that incorporates additional randomness into the selection of allocation rules, leading to the concepts of tOCRS and tCRS. Treating the relaxation (tOCRS or tCRS) as an offline oracle, the algorithm proceeds by querying the oracle to ensure that the selected allocation is feasible. It then decides whether to allocate or not using a second layer of Bernoulli sampling.

The paper establishes theoretical guarantees for the proposed algorithm by proving both the approximation ratio and the validity of the BIC and BIR properties. Finally, the paper concludes by interpreting these theoretical results in the context of the three motivating applications discussed earlier, demonstrating the practical relevance of the framework.

**Questions:**

Please see the above weakness section.

**Ethical Concerns:**

["NO or VERY MINOR ethics concerns only"]

**Final Justification:**

I thank the authors for the responses. I think the implementation of the algorithm requires more careful thoughts and efforts. Maybe a concrete experimental plan or even a very simple toy example would be helpful. As a result, I will keep my score as it is.

**Quality:**

3

**Strengths And Weaknesses:**

Strength

- The paper is clearly written and enjoyable to read; a quite complicated problem is very well laid out

- The algorithm is elegant and easy to interpret

- The paper theoretically establishes the feasibility, and the approximation ratio of the algorithm

- The application in Section 5 is very clear in supporting the insights of the theoretical results

Weakness

- I would appreciate highlighting the novelty more in the first few pages of the paper; the paper has not elaborated on the novel sections until page 6. I understand that it is for clarity, and it might be great if these two could be balanced. For example, the framework for
tCRSs could be kept in the main body rather than in the appendix, since this is something new.

- The description about flipping a coin with probability $p^{*}_{i,j}(v_i)$ without knowledge of this probability term might need further justification. In particular, why could this way get rid of the dependency on the knowledge of this probability?

- The construction procedures of tOCRs is only illustrated for VH family or more specifically, Knapsack constraints.

- Proposition 1 and 2 does not tell us how to implement such tOCRs given the constraints, which is of great importance to the algorithm, since part of the key factors of Algorithm 1 is the existence of such tOCRs.

- I guess the main concern Is whether one can really implement this algorithm from scratch down to earth, based on the pseudo code. It would be very helpful if there is a case study containing certain numerical results to validate the practical use case of the proposed algorithm.

- It is unclear whether the approximation ratio is ratio; a comparison with the related work would be helpful.

- Would it be possible to remove the assumption that the valuations of different agents are independent of one another? Given the era of digitalization, it is possible that agents can communicate or gather information about one another.

- It would be helpful to include a conclusion and future work section, summarizing the paper and inspiring subsequent study along this direction.

---

> ### Author Rebuttal · Authors · 2025-07-30
>
> Thank you for your review. We comment on some of the weaknesses you point out in your review.
>
> >  The description about flipping a coin with probability $p^*_{i,j}(v_i)$  without knowledge of this probability term might need further justification. In particular, why could this way get rid of the dependency on the knowledge of this probability?
>
> (Markdown was having issues with $p^*_{i,j}(v_i)$, so we will be using $p$ in this response)
>
> In order for the incentive constraints to be satisfied, we cannot afford approximations when using randomness --- we need to have a perfect handle on all the randomness we use. What we need is to output 1 with probability $f(p)$, for some function $f(.)$.
>
> Due to the nature of our setup, we do not have exact knowledge of $p$, in the sense that it is not explicitly written down in memory. We do, however, have access to a Bernoulli oracle that outputs 1 with probability $p$ and 0 otherwise. While one could sample repeatedly from this oracle to approximate $p$ up to any desired accuracy, this would still not be good enough: we would only get approximate incentive compatibility. The Bernoulli factory problem studies precisely the question we need: is it possible to implement a coin with bias $f(q)$ given black-box access to independent samples from a Bernoulli distribution with bias $q$? To efficiently implement our mechanisms while preserving exact truthfulness, we use Bernoulli factories for division.
>
> > The construction procedures of tOCRs is only illustrated for VH family or more specifically, Knapsack constraints.
>
> We believe that this question stems from a misunderstanding. First, Knapsack constraints are not in the VH family. Our tOCRSs for Knapsack and Multi-Choice Knapsack are new. Second, Theorem 2 allows us to efficiently construct numerous tCRSs and tOCRSs from existing CRSs and OCRSs, including matchings, matroids, and so on. These constructions are also computationally efficient.
>
> > Proposition 1 and 2 does not tell us how to implement such tOCRs given the constraints, which is of great importance to the algorithm, since part of the key factors of Algorithm 1 is the existence of such tOCRs.
>
> This also appears to be a misunderstanding. Theorems 3 and 4 show that certain tOCRSs (for Knapsack and Multi-Choice Knapsack) *exist*. Propositions 1 and 2, respectively, give efficient implementations for these tOCRSs (with only a tiny loss in the guarantee). So, using Propositions 1 and 2, we can get an efficient implementation of Algorithm 1.
>
> >  I guess the main concern is whether one can really implement this algorithm from scratch down to earth, based on the pseudo code. It would be very helpful if there is a case study containing certain numerical results to validate the practical use case of the proposed algorithm.
>
> Yes, all our algorithms can be efficiently implemented from scratch. The input is a feasible in expectation interim rule, and the output is a mechanism (and even a sequential mechanism in most cases). We show how to efficiently implement every step; see Section 5 for the detailed steps for each application (we can also revise Application 2 to explicitly point to Propositions 1 and 2 if that avoids confusion). If the designer does not already have the interim rule, but only the agent distributions and feasibility constraints, the interim rule we need can be computed using LP1. LP1 can be solved efficiently for many feasibility constraints using the framework of Cai et al.; we note that our LP1 is, in fact, a lot simpler than the LPs Cai et al. need to solve, since we only need feasibility in expectation, not ex-post.
>
> > It is unclear whether the approximation ratio is ratio; a comparison with the related work would be helpful.
>
> We are unsure what this question is asking. If the reviewer is asking how our approximation ratios compare to existing work, the framework of Cai et al. can be used to *compute* the optimal mechanism exactly. In contrast, the literature on *simple* approximately optimal mechanisms has considered much more restricted feasibility constraints (e.g., each item can be allocated to at most one agent), and typically non-sequential mechanisms. Our mechanisms are more complex than the mechanisms championed in this literature; in exchange, we give sequential mechanisms, and our work can facilitate much more general constraints, previously out of reach for this line of work.
>
> > Would it be possible to remove the assumption that the valuations of different agents are independent of one another? Given the era of digitalization, it is possible that agents can communicate or gather information about one another.
>
> This is a great question. Unfortunately, in mechanism design, broadly speaking, we only know how to get revenue-optimal, or even approximately optimal, mechanisms under the independence assumption. There are known barriers (e.g., computational barriers) for the case of correlated types. That said, there are some important exceptions (and “twists” to the standard model) where correlation can be handled; adapting our framework to work here is an interesting direction for future work.

---

> > ### Comment · Reviewer_y7Bn · 2025-08-07
> >
> > I thank the authors for the responses. I think the implementation of the algorithm requires more careful thoughts and efforts. Maybe a concrete experimental plan or even a very simple toy example would be helpful.

---

### Decision · Program_Chairs · 2025-09-17

**Decision:**

Accept (poster)

**Comment:**

This theoretical paper introduced a framework to design revenue maximizing mechanisms under feasibility constraints. The paper presented a few applications of their framework in various special cases. Reviewers found the problem important, the theoretical results solid, nontrivial, and useful from a theoretical point of view. The main reservation is the hardness to implement in practice. Some reviewers found the paper hard to read while other found it well-written, especially the proofs. The overall sentiment slightly went up after the discussions and rebuttal.